# BENEFIT OF INTERPOLATION IN NEAREST NEIGHBOR ALGORITHMS

## ABSTRACT

The over-parameterized models attract much attention in the era of data science and deep learning. It is empirically observed that although these models, e.g. deep neural networks, over-fit the training data, they can still achieve small testing error, and sometimes even *outperform* traditional algorithms which are designed to avoid over-fitting. The major goal of this work is to sharply quantify the benefit of data interpolation in the context of nearest neighbors (NN) algorithm. Specifically, we consider a class of interpolated weighting schemes and then carefully characterize their asymptotic performances. Our analysis reveals a U-shaped performance curve with respect to the level of data interpolation, and proves that a mild degree of data interpolation *strictly* improves the prediction accuracy and statistical stability over those of the (un-interpolated) optimal $k$NN algorithm. This theoretically justifies (predicts) the existence of the second U-shaped curve in the recently discovered double descent phenomenon. Note that our goal in this study is not to promote the use of interpolated-NN method, but to obtain theoretical insights on data interpolation inspired by the aforementioned phenomenon.

## 1 INTRODUCTION

Classical statistical learning theory believes that over-fitting deteriorates prediction performance: when the model complexity is beyond necessity, the testing error must be huge. Therefore, various techniques have been proposed in literature to avoid over-fitting, such as early stopping, dropout and cross validation. However, recent experiments reveal that even with over-fitting, many learning algorithms still achieve small generalization error. For instances, Wyner et al. (2017) explored the over-fitting in AdaBoost and random forest algorithms; Belkin et al. (2019a) discovered a double descent phenomenon in random forest and neural network: with growing model complexity, testing performance firstly follows a (conventional) U-shaped curve, and as the level of overfitting increases, a second descent or even a second U-shaped testing performance curve occurs.

To theoretically understand the effect of over-fitting or data interpolation, Du & Lee (2018); Du et al. (2019; 2018); Arora et al. (2018; 2019); Xie et al. (2017) analyzed how to train neural networks under over-parametrization, and why over-fitting does not jeopardize the testing performance; Belkin et al. (2019c) constructed a Nadaraya-Watson kernel regression estimator which perfectly fits training data but is still minimax rate optimal; Belkin et al. (2018) and Xing et al. (2018) studied the rate of convergence of interpolated nearest neighbor algorithm (interpolated-NN); Belkin et al. (2019b); Bartlett et al. (2019) quantified the prediction MSE of the linear least squared estimator when the data dimension is larger than sample size and the training loss attains zero. Similar analysis is also conducted by Hastie et al. (2019) for two-layer neural network models with a fixed first layer.

In this work, we aim to provide theoretical reasons on whether, when and why the interpolated-NN performs better than the optimal $k$NN, by some *sharp* analysis. The classical $k$NN algorithm for either regression or classification is known to be rate-minimax under mild conditions (Chaudhuri & Dasgupta (2014)), say $k$ diverges propoerly. However, can such a simple and versatile algorithm still benefit from intentional over-fitting? We first demonstrate some empirical evidence below.

Belkin et al. (2018) designed an interpolated weighting scheme as follows:

$$\widehat{y}(x) = \frac{\sum_{i=1}^{k} \|x_{(i)} - x\|^{-\gamma} y(x_{(i)})}{\sum_{i=1}^{k} \|x_{(i)} - x\|^{-\gamma}}, \tag{1}$$

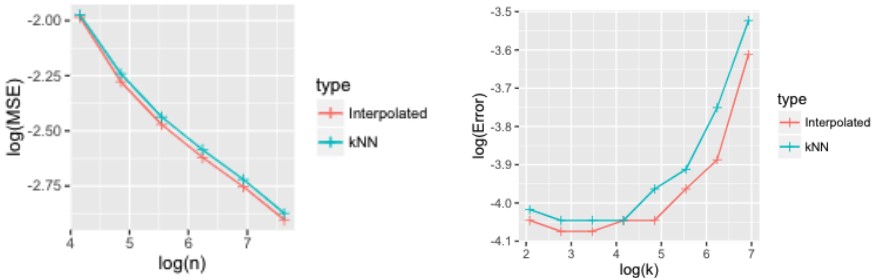

Figure 1: Regression in Simulation (Left) and Classification in Real Data (Right), Comparison between $k$NN and Interpolated-NN (with logarithm weight) in Xing et al. (2018).

where $x_{(i)}$ is the $i$-th closest neighbor to $x$ with the corresponding label $y(x_{(i)})$. The parameter $\gamma \geq 0$ controls the level of interpolation: with a larger $\gamma > 0$, the algorithm will put more weights on the closer neighbors. In particular when $\gamma = 0$ or $\gamma = \infty$, interpolated-NN reduces to $k$NN or 1-NN, respectively. Belkin et al. (2018) showed that such an interpolated estimator is rate minimax in the regression setup, but suboptimal in the setting of binary classification. Later, Xing et al. (2018) obtained the minimax rate of classification by adopting a slightly different interpolating kernel. What is indeed more interesting is the preliminary numerical analysis (see Figure 1) conducted in the aforementioned paper, which demonstrates that interpolated-NN is even better than the rate minimax $k$NN in terms of MSE (regression) or mis-classification rate (classification). This observation asks for deeper theoretical exploration beyond the rate of convergence. A reasonable doubt is that the interpolated-NN may possess a smaller multiplicative constant for its rate of convergence, which may be used to study the generalization ability within the "over-parametrized regime."

In this study, we will theoretically compare the minimax optimal $k$NN and the interpolated-NN (under (1)) in terms of their multiplicative constants. On the one hand, we show that under proper smooth conditions, the multiplicative constant of interpolated-NN, as a function of interpolation level $\gamma$, is U-shaped. As a consequence, interpolation indeed leads to more accurate and stable performance when the interpolation level $\gamma \in (0, \gamma_d)$ for some $\gamma_d > 0$ only depending on the data dimension $d$. The amount of benefit (i.e., the "performance ratio" defined in Section 2) follows exactly the same asymptotic pattern for both regression and classification tasks. In addition, the gain from interpolation diminishes as the dimension $d$ grows to infinity, i.e. high dimensional data benefit less from data interpolation. We also want to point out that there still exist other "non-interpolating" weighting schemes, such as OWNN, which can achieve an even better performance; see Section 3.4. More subtle results are summarized in the figure below.

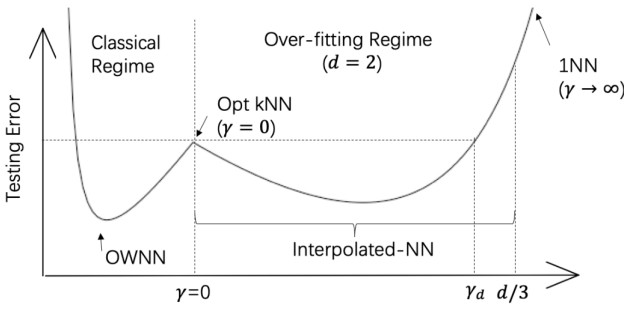

Figure 2: Testing Error with Model Complexity / Interpolation Level. The curve within over-fitting regime is depicted based on the theoretical pattern of performance ratio given $d = 2$.

From Figure 2, we theoretically justify (predict) the existence of the U-shaped curve within the "over-fitting regime" of the recently discovered double descent phenomenon by Belkin et al.

(2019a;b). As complementary to Belkin et al. (2018); Xing et al. (2018), we further show in appendix that interpolated-NN reaches optimal rate for both regression and classification under *more general* $(\alpha, \beta)$-smoothness conditions in Section F.

In the end, we want to emphasize that our goal here is not to promote the practical use of this interpolation method given that $k$NN is more user-friendly. Rather, the interpolated-NN algorithm is used to precisely describe the role of interpolation in generalization ability so that more solid theoretical arguments can be made for the very interesting double descent phenomenon, especially in the over-fitting regime.

## 2 INTERPOLATION IN NEAREST NEIGHBORS ALGORITHM

In this section, we review the interpolated-NN algorithm introduced by Belkin et al. (2018) in more details. Given $x$, we define $R_{k+1}(x)$ to be the distance between $x$ and its $(k+1)$th nearest neighbor. W.O.L.G, we let $X_1$ to $X_k$ denote the (unsorted) $k$ nearest neighbors of $x$, and let $\{R_i(x)\}_{i=1}^k$ to be distances between $x$ and $X_i$. Thus, based on the same argument used in Chaudhuri & Dasgupta (2014) and Belkin et al. (2018), conditional on $R_{k+1}(x)$, $X_1$ to $X_k$ are iid variables whose support is a ball centered at $x$ with radius $R_{k+1}(x)$, and as a consequence, $R_1(x)$ to $R_k(x)$ are conditionally independent given $R_{k+1}(x)$ as well. Note that when no confusion is caused, we will write $R_i(x)$ as $R_i$. Thus, the weights of the $k$ neighbors are defined as

$$W_i = \frac{R_i^{-\gamma}}{\sum_{j=1}^k R_j^{-\gamma}} = \frac{(R_i/R_{k+1})^{-\gamma}}{\sum_{j=1}^k (R_j/R_{k+1})^{-\gamma}},$$

for $i = 1, \ldots, k$ and some $\gamma \geq 0$.

For regression models, denote $\eta(x)$ as the target function, and $Y(x) = \eta(x) + \epsilon(x)$ where $\epsilon(x)$ is an independent zero-mean noise with $\mathbb{E}\epsilon(x)^2 := \sigma^2(x)$. The regression estimator at $x$ is thus

$$\widehat{\eta}_{k,n,\gamma}(x) = \sum_{i=1}^k W_i Y_i.$$

For binary classification, denote $\eta(x) = P(Y = 1|X = x)$, with $g(x) = 1\{\eta(x) > 1/2\}$ as the Bayes estimator. The interpolated-NN classifier is defined as

$$\widehat{g}_{k,n,\gamma}(x) = \begin{cases} 1 & \sum_{i=1}^k W_i Y_i > 1/2 \\ 0 & \sum_{i=1}^k W_i Y_i \leq 1/2 \end{cases}.$$

As discussed previously, the parameter $\gamma$ controls the level of interpolation: a larger value of $\gamma$ leads to a higher degree of data interpolation.

We adopt the conventional measures to evaluate the theoretical performance of interpolated-NN given a new test data $(X, Y)$:

$$\text{Regression:} \quad \text{MSE}(k, n, \gamma) = \mathbb{E}((\widehat{\eta}_{k,n,\gamma}(X) - \eta(X))^2).$$

$$\text{Classification:} \quad \text{Regret}(k, n, \gamma) = P(\widehat{g}_{k,n,\gamma}(X) \neq Y) - P(g(X) \neq Y).$$

## 3 QUANTIFICATION OF INTERPOLATION EFFECT

### 3.1 MODEL ASSUMPTIONS

Recent works by Belkin et al. (2018) and Xing et al. (2018) confirm the rate optimality of MSE and regret for interpolated-NN under mild interpolation conditions. Two deeper questions (hinted by Figure 1) we would like to address are whether and how interpolation strictly benefits NN algorithm, and whether interpolation affects regression and classification in the same manner.

To facilitate our theoretical investigation, we impose the following assumptions:

A.1 $X$ is a $d$-dimensional random variable on a compact set $\mathcal{X}$ with boundary $\partial \mathcal{X}$.

A.2 For classification, $\mathcal{S} = \{x|\eta(x) = 1/2\}$ is non-empty.

A.3 $d - 3\gamma \geq C > 0$ for some constant $C$.

A.4 For classification, $\eta$ is continuous in some open set containing $\mathcal{X}$. The third-order derivative of $\eta$ is bounded when $\eta(x) = 1/2 \pm c_0$ for a small constant $c_0 > 0$. The gradient $\dot{\eta}(x) \neq 0$ when $\eta(x) = 1/2$, and with restriction on $x \in \partial\mathcal{X}$, $\dot{\partial}\eta(x) \neq 0$ if $\eta(x) = 1/2$ and $x \in \partial\mathcal{X}$.

A.5 For classification, density of $X$, denoted as $f$, is twice differentiable and finite.

A.6 For regression, the third-order derivative of $\eta$ is bounded for all $x$.

A.7 For regression, $\sup_{x \in \mathcal{X}} \sigma^2(x)$ is finite and $\sigma^2(x)$ has finite first-order derivative in $x$.

The above assumptions (except A.3) are mostly derived from the framework established by Samworth et al. (2012). Note that the additional smoothness required in $\eta$ and $f$ is needed to facilitate the asymptotic study of interpolation weighting scheme. We also want to point out that these assumptions are generally stronger than those used in Chaudhuri & Dasgupta (2014), but necessary to figure out the multiplicative constant. Further discussions regarding the conditions can be found in Remark 3 in appendix.

## 3.2 MAIN THEOREM

The following theorem examines the asymptotic performance ratios of MSE and Regret between interpolated-NN and $k$NN and discovers that these ratios (under their respective optimal choice of $k$) converges to a function of $(d, \gamma)$ only. In particular, a U-shaped curve is revealed where the ratio is smaller than 1 when $\gamma \in (0, \gamma_d)$ for some $\gamma_d < d/3$. Define the minimum MSE and Regret over $k \in (n^\beta, n^{1-4\beta/d})$ as follows:

$$\text{MSE}(\gamma, n) = \min_{k \in (n^\beta, n^{1-4\beta/d})} \text{MSE}(k, n, \gamma) \quad \text{and} \quad \text{Regret}(\gamma, n) = \min_{k \in (n^\beta, n^{1-4\beta/d})} \text{Regret}(k, n, \gamma).$$

Theorem 1 asymptotically compares the interpolated-NN and $k$NN, i.e., $\gamma = 0$, in terms of the above measures. Interestingly, it turns out that the performance ratio, defined as

$$\text{PR}(d, \gamma) := \left(1 + \frac{\gamma^2}{d(d - 2\gamma)}\right)^{\frac{4}{d+4}} \left(\frac{(d - \gamma)^2}{(d + 2 - \gamma)^2} \frac{(d + 2)^2}{d^2}\right)^{\frac{d}{d+4}},$$

is a function of $d$ and $\gamma$ only, independent of the underlying data distribution. Note that $\text{PR}(d, \gamma)$ is just the ratio of multiplicative constants before the minimax rate of interpolated-NN and $k$NN.

**Theorem 1** *For regression, suppose that assumptions A.1, A.3, A.6, and A.7 hold. For classification, under A.1 to A.5, for any $\gamma \in [0, d/3)$,*

$$\frac{MSE(n, \gamma)}{MSE(n, 0)} \to PR(d, \gamma), \quad and \quad \frac{Regret(n, \gamma)}{Regret(n, 0)} \to PR(d, \gamma), \quad as\ n \to \infty.$$

*Note that $MSE(n, 0)$/$Regret(n, 0)$ is the optimum MSE/Regret for kNN.*

The proof Theorem 1 are postponed to appendix (Section D).

When $k$ can be chosen adaptively based on $\gamma$, we can address the second question that interpolation affects regression and classification in exactly the same manner through $\text{PR}(d, \gamma)$. In particular, this ratio exhibits an interesting U-shape of $\gamma$ for any fixed $d$. Specifically, as $\gamma$ increases from 0, $\text{PR}(d, \gamma)$ first decreases from 1 and then increases above 1; see Figures 2 and 3. Therefore, within the range $(0, \gamma_d)$ for some $\gamma_d$ only depending on dimension $d$, $\text{PR}(d, \gamma) < 1$, that is, the interpolated-NN is *strictly* better than the $k$NN. Given the imposed condition that $\gamma < d/3$, Some further calculations reveal that $\gamma_d < d/3$ when $d \leq 3$; $\gamma_d = d/3$ when $d \geq 4$.

**Remark 1** *It is easy to show that $\lim_{d \to \infty}[\min_{\gamma < d/3} PR(d, \gamma)] = 1$. This indicates that high dimensional model benefits less from interpolation, or said differently, high dimensional model is less affected by data interpolation. This phenomenon can be explained by the fact that, as $d$ increases, $R_i/R_{k+1} \to 1$ due to high dimensional geometry.*

**Remark 2** *The optimum $k$, which leads to the best MSE/regret, depends on the interpolation level $\gamma$. Thus, we denote it as $k_\gamma$. As shown in the appendix, $k_\gamma \asymp k_0(:= k_{\gamma=0}) \asymp n^{\frac{4}{d+4}}$, but $k_\gamma/k_0 > 1$ for $\gamma > 0$, i.e., interpolated-NN needs to employ slightly more neighbors to achieve the best performance. Empirical support for this finding can be found in Section A of appendix. If we insist using the same $k \asymp n^{\frac{4}{d+4}}$ for interpolated-NN and kNN, we can still verify that $MSE(k, n, \gamma)/MSE(k, n, 0) < 1$ and $Regret(k, n, \gamma)/Regret(k, n, 0) < 1$, when $\gamma < \gamma_x$ for some $\gamma_x$ depending on the distribution of $X$ and $\eta$.*

### 3.3  STATISTICAL STABILITY

In this section, we will explore how the interpolation affects the statistical stability of nearest neighbor classification algorithms. This is beyond the generalization results obtained in Section 3.2. In short, if we choose the best $k$ in $k$NN and apply it to the interpolated-NN, then $k$NN will be more stable; otherwise, the interpolated-NN will be more stable for $\gamma \in (0, \gamma_d)$ if the $k$ is allowed to be chosen separately and optimally based on $\gamma$.

For a stable classification method, it is expected that with high probability, the classifier can yield the same prediction when being trained by different data sets sampled from the same population. As a result, Sun et al. (2016) introduced a type of statistical stability, classification instability (CIS), which is different from the algorithmic stability in the literature (Bousquet & Elisseeff, 2002). Denote $\mathcal{D}_1$ and $\mathcal{D}_2$ as two i.i.d. training sets with the same sample size $n$. The CIS is defined as:

$$\text{CIS}_{k,n}(\gamma) = P_{\mathcal{D}_1, \mathcal{D}_2, X}\left(\widehat{g}_{k,n,\gamma}(x, \mathcal{D}_1) \neq \widehat{g}_{k,n,\gamma}(x, \mathcal{D}_2)\right).$$

Hence, a larger value of CIS indicates that the classifier is less statistically stable. In practice, we need to take into account of mis-classification rate and classification instability at the same time. Therefore, we are interested in comparing the stability between interpolated-NN and $k$-NN only when the regrets of both algorithms reach their optimal performance under respective optimal $k$ choices.

Theorem 2 below illustrates how CIS is affected by interpolation through $k$, $n$ and $\gamma$.

**Theorem 2** *Under the conditions in Theorem 1, the CIS of interpolated-NN is derived as*

$$CIS_{k,n}(\gamma) = \frac{B_1}{\sqrt{\pi}} \frac{1}{\sqrt{k}} \mathbb{E}s_{k,n,\gamma}(X) + o.$$

The proof of Theorem 2 is postponed to Section E in appendix.

Similarly, Corollary 3 asymptotically compares CIS between interpolated-NN and $k$NN.

**Corollary 3** *Following the conditions in Theorem 2, when the same $k$ value is used for kNN and interpolated-NN, then as $n \to \infty$,*

$$\frac{CIS_{k,n}(\gamma)}{CIS_{k,n}(0)} > 1.$$

*On the other hand, if we choose optimum $k$'s for kNN and interpolated-NN respectively, i.e. $k_\gamma = \arg\min_k Regret(k, n, \gamma)$, when $n \to \infty$, we have*

$$\left(\frac{CIS_{k_\gamma,n}(\gamma)}{CIS_{k_0,n}(0)}\right)^2 \to PR(d, \gamma).$$

*Therefore, when $\gamma \in (0, \gamma_d)$, interpolated-NN with optimal $k$ has higher accuracy and stability than $k$-NN at the same time.*

From Corollary 3, the interpolated-NN is not as stable as $k$NN if the same number of neighbors is used in both algorithms. However, this is not the case if an optimal $k$ is chosen separately. An intuitive explanation is that, under the same $k$, $k$NN has a smaller variance (more stable) given equal weights for all $k$ neighbors; on the other hand, by choosing an optimum $k$, the interpolated-NN can achieve a much smaller bias, which offsets its performance lost in variance through enlarging $k$.

### 3.4 CONNECTION WITH OWNN AND DOUBLE DESCENT PHENOMENON

Samworth et al. (2012) firstly worked out a general form of regret using a rank-based weighting scheme, and proposed the optimally weighted nearest neighbors algorithm (OWNN). The OWNN is the best nearest neighbors algorithm in terms of minimizing MSE for regression (and Regret for classification), when the weights of neighbors are only rank-based.

Combining Theorem 1 with Samworth et al. (2012), we can further compare the interpolated-NN against OWNN as follows:

$$\frac{R(n, \text{OWNN})}{R(n, \gamma)} \to 2^{\frac{4}{d+4}} \left(\frac{d+2}{d+4}\right)^{\frac{2d+4}{d+4}} \left(1 + \frac{\gamma^2}{d(d-2\gamma)}\right)^{-\frac{4}{d+4}} \left(\frac{(d-\gamma)^2}{(d+2-\gamma)^2} \frac{(d+2)^2}{d^2}\right)^{-\frac{d}{d+4}},$$

which is always smaller than 1 (just by definition). Here $R(n, \text{OWNN})$ denotes the MSE/Regret of OWNN given its optimum $k$, and $R(n, \gamma)$ denotes the one of interpolated-NN given its own optimum $k$ choice. It is interesting to note from the above ratio that that the advantage of OWNN is only reflected at the level of multiplicative constant, and further that the ratio converges to 1 as $d$ diverges (just as the case of PR$(d, \gamma)$; see Remark 1). Thus, under ultra high dimensional setting, the performance differences among $k$NN, interpolated-NN and OWNN are almost negligible even at the multiplicative constant level.

We first describe the framework of the recently discovered double descent phenomenon (e.g., Belkin et al., 2019a;b), and then comment our contributions (summarized in Figure 2) to it in the context of nearest neighbor algorithm. Specifically, within the "classical regime" where exact data interpolation is impossible, the testing performance curve is the usual U-shape w.r.t. model complexity; once the model complexity is beyond a critical point it thus enters the "over-fitting regime," the testing performance will start to decrease again as severeness of data interpolation increases, which is so-call "double descend".

In the context of nearest neighbors algorithms, different weighting schemes may be viewed as a surrogate of modeling complexity. For OWNN, though it allocates more weights on closer neighbors, while none of the weights exceeds $(1 + d/2)/k$. Thus, OWNN is never an interpolation weighting scheme. From this aspect, $k$-NN and OWNN both belong to the "classical regime," while interpolated-NN is within the "over-fitting regime." In particular, the testing performance of OWNN reaches the minimum point of the U-shaped curve inside the "classical regime." Deviation from this optimal choice of weight leads to the increase of the MSE/Regret within this "classical regime." After the "over-fitting regime" is reached by the interpolated-NN, say from $\gamma = 0$ in Figure 2, the MSE/Regret decreases as the interpolation level $\gamma$ increases within the range $(0, \gamma_d)$ and ascends again when $\gamma > \gamma_d$ (if the dimension allows $\gamma_d < d/3$), forming the second U-shaped curve in Figure 2. Therefore, we obtain an overall W-shaped performance curve *with theoretical guarantee*, which coincides the empirical finding of Belkin et al. (2019b) for over-parametrized linear models.

## 4 NUMERICAL EXPERIMENTS

In this section, we will present several simulation studies to justify our theoretical discoveries for regression, classification and stability performances of the interpolated-NN algorithm, together with some real data analysis.

### 4.1 SIMULATIONS

We aim to estimate the performance ratio curve by data simulation and compare it with the theoretical curve PR$(d, \gamma)$. The second simulation setting in Samworth et al. (2012) is adopted here. Specifically, the joint distribution of $(X = (X^{(i)})_{i=1}^d, Y)$ follows $P(Y = 1) = P(Y = 0) = 1/2$, $f(X|Y = 0) = \prod_i [\frac{1}{2}\phi(X^{(i)}; 0, 1) + \frac{1}{2}\phi(X^{(i)}; 3, 2)]$ and $f(X|Y = 1) = \prod_i [\frac{1}{2}\phi(X^{(i)}; 1.5, 1) + \frac{1}{2}\phi(X^{(i)}; 4.5, 2)]$, where $\phi(\cdot; \mu, \sigma^2)$ denotes the density of $N(\mu, \sigma^2)$. The sample size $n = 2^{10}$ and dimension $d = 2, 5$. The interpolated-NN regressor and classifier were implemented under different choices of $\gamma/d = 0, 0.05, 0.1, \ldots, 0.35$ and $k = 1, \ldots, n$. For regression, the MSE was estimated based on 100 repetitions, and for classification, the Regret was based on 500 repetitions.

When $n = 1024$, the Regret/MSE ratio for different $\gamma/d$ is shown in Figure 3. Here Regret ratio is defined by Regret$(n, \gamma)$/Regret$(n, 0)$, the MSE ratio is defined by MSE$(n, \gamma)$/MSE$(n, 0)$. The

trends for theoretical value and simulation value are mostly close. The small difference is mostly caused by the small order terms in the asymptotic result and shall vanish if larger $n$ is used. Note that $\gamma/d = 0.35$ is outside our theoretical range $\gamma/d < 1/3$, but the performance is still reasonable in our numerical experiment.

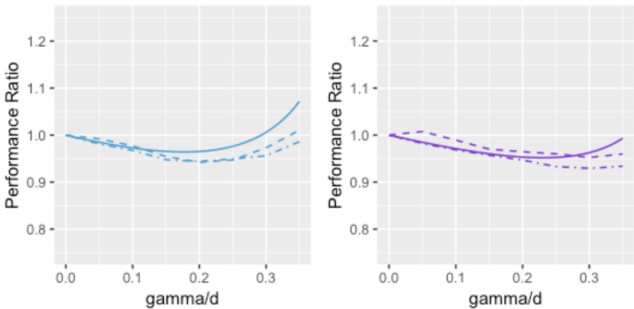

Figure 3: Performance ratio curve for $n = 1024$ and $d = 2$ (left) and 5 (right), solid line is theoretical value $\mathrm{PR}(d, \gamma)$, dashed line is simulation results for Regret ratio, dashed line with dots is simulation results for MSE ratio.

We further estimate CIS by training two classifiers based on two different simulated data sets of 1024 samples. The CIS was estimated by calculating the proportion of testing samples that have different prediction labels, that is

$$\widehat{\mathrm{CIS}}(\gamma) = \frac{1}{n} \sum_{i=1}^{n} 1(\widehat{g}(x_i, \mathcal{D}_1) \neq \widehat{g}(x_i, \mathcal{D}_2)).$$

The CIS result is shown in Figure 4. One can see that when $\gamma$ is small, the simulated CIS ratio

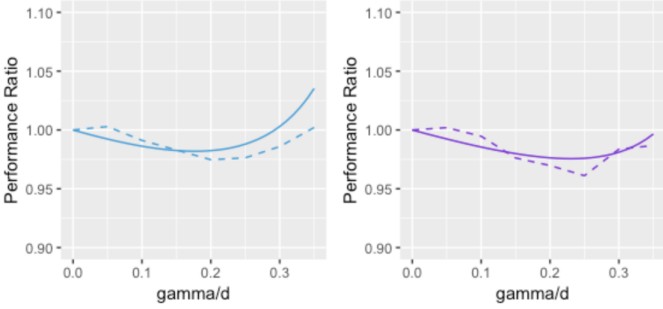

Figure 4: Comparison between theoretical CIS ratio $\sqrt{\mathrm{PR}(d, \gamma)}$ (solid line) and simulated CIS ratio (dashed line), under $d = 2$ (left) and 5 (right).

decreases in a similar manner as the asymptotic value, while simulated value will increase when $\gamma$ gets larger. This pattern is the same as the theoretical result predicted in Theorem 2.

An additional experiment is postponed to appendix, which shows how the MSE and optimum $k$ changes in $\gamma$ and $n$ where $n = 2^i$ for $i = 6, \ldots, 10$.

## 4.2 REAL DATA ANALYSIS

In real data experiment, we compare the classification accuracy of interpolated-NN with $k$NN.

Five data sets were considered in this experiment. The data set HTRU2 from Lyon et al. (2016) uses 17,897 samples with 8 continuous attributes to classify pulsar candidates. The data set Abalone contains 4,176 samples with 7 attributes. Following Wang et al. (2018), we predict whether the number of rings is greater than 10. The data set Credit (Yeh & Lien, 2009) has 30,000 samples with 23 attributes, and predicts whether the payment will be default in the next month given the current

| Data | $d$ | Error ($\gamma = 0$) | Error (best $\gamma/d$) | best $\gamma/d$ |
|------|-----|----------------------|-------------------------|-----------------|
| Abalone | 7 | 0.22239 | 0.22007 | 0.3 |
| HTRU2 | 8 | 0.02315 | 0.0226 | 0.2 |
| Credit | 23 | 0.1933 | 0.19287 | 0.05 |
| Digit | 64 | 0.01745 | 0.01543 | 0.25 |
| MNIST | 784 | 0.04966 | 0.04656 | 0.05 |

Table 1: Prediction Error of $k$NN, interpolated-NN under the best choice of $\gamma$, together with the value of the best $\gamma$ for interpolated-NN.

payment information. The built-in data set of digits in *sklearn* (Pedregosa et al., 2011) contains 1,797 samples of $8{\times}8$ images. For images in MNIST are $26 \times 26$, we will use part of it in our experiment. Both the data set of digit and MNIST have ten classes. Here for binary classification we group 0 to 4 as the first class and 5 to 9 as the second class.

For each data set, a proportion of data is used for training and the rest is reserved to test the accuracy of the trained classifiers. For Abalone, HTRU2, Credit and Digit, we use 25% data as training data and 75% as testing data. For MNIST, we use randomly choosen 2000 samples as training data and 1000 as testing data, which is sufficient for our comparison. The above experiment is repeated for 50 times and the average testing error rate is summarized in Table 1. For all data sets, the testing error of interpolated-NN (column "best $\gamma/d$") is always smaller than the $k$NN(column "$\gamma = 0$"), which verifies that nearest neighbor algorithm actually benefits from interpolation.

## 5 CONCLUSION

Our work precisely quantifies how data interpolation affects the performance of nearest neighbor algorithms beyond the rate of convergence. We find that for both regression and classification problems, the asymptotic performance ratios between interpolated-NN and $k$NN converge to the same value, which depends on $d$ and $\gamma$ only. More importantly, when the interpolation level $\gamma/d$ is within a reasonable range, the interpolated-NN is strictly better than $k$NN as it has a smaller multiplicative constant of the convergence rate, and it has a more stable prediction performance as well.

Classical learning framework opposes data interpolation as it believes that over-fitting means fitting the random noise rather than the model structures. However, in the interpolated-NN, the weight degenerating occurs only on a nearly-zero-measure set, and thus there is only "local over-fitting", which may not hurt the overall rate of convergence. Technically, through balancing the variance and bias, data interpolation can possibly improve the overall performance. And our work essentially quantify such a bias-variance balance in a very precise way. It is of great interest to investigate how our theoretical insights can be carried over to the real deep neural networks, leading to a more complete picture of double descent phenomenon.

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

## A    ADDITIONAL NUMERICAL EXPERIMENT

In this experiment, instead of taking $n = 2^{10}$ only, we take $n = 2^i$ for $i = 6, \ldots, 10$ to see how the performance ratio and optimum $k$ changes in $\gamma$ for different $n$'s. The phenomenon for classification is similar as regression so we only present regression. Figure 5 summarizes the change of MSE and optimum choice of $k$ with respect to different choices of $n$ and $\gamma/d$, when $d = 2$. The plot corresponding to $d = 5$ is quite similar hence omitted here. This plot shows that with respect to the increase of $n$, interpolated-NN converges in the same rate as $k$NN, and interpolated-NN generally requires larger $k$ than $k$-NN.

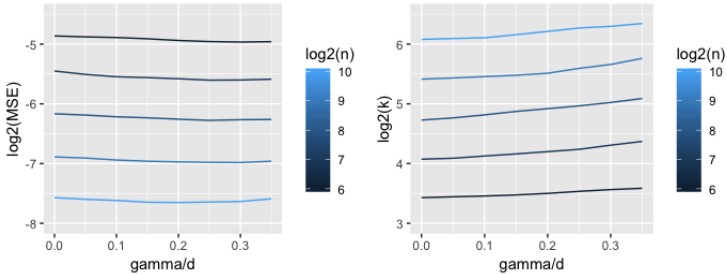

Figure 5: Change of MSE and optimal $k$

## B    PRELIMINARY PROPOSITION

This section provides an useful result when integrating c.d.f:

**Proposition 4** *From Lemma S.1 in Sun et al. (2016), we have for any distribution function $G$,*

$$
\int_{\mathbb{R}} [G(-bu - a) - 1_{\{u<0\}}] du = -\frac{1}{b} \left\{ a + \int_{\mathbb{R}} t\, dG(t) \right\},
$$
$$
\int_{\mathbb{R}} u[G(-bu - a) - 1_{\{u<0\}}] du = \frac{1}{b^2} \left\{ \frac{a^2}{2} + \frac{1}{2} \int_{\mathbb{R}} t^2\, dG(t) + a \int_{\mathbb{R}} t\, dG(t) \right\}.
$$

## C    PROOF OF A PRELIMINARY THEOREM FOR THEOREM 1

Define $P_1$ and $P_2$ (density as $f_1$, $f_2$) as the conditional distributions of $X$ given $Y = 0, 1$ respectively, and $\pi_1, \pi_2$ are the marginal probability $P(Y = 0)$ and $P(Y = 1)$, then take $P = \pi_1 P_2 + \pi_2 P_2$, $\bar{P} = \pi_1 P_1 - \pi_2 P_2$, $f(x) = \pi_1 f_1(x) + \pi_2 f_2(x)$, and also denote $\Psi(x) = \pi_1 f_1(x) - \pi_2 f_2(x)$.

**Theorem 5** *For regression, suppose that assumptions A.1, A.3, A.6, and A.7 hold. If $k$ satisfies $n^\beta \le k \le n^{1-4\beta/d}$ for some $\beta > 0$, we have* [1]

$$
\begin{aligned}
MSE(k,n,\gamma) &= k\mathbb{E}\left[\frac{(R_1/R_{k+1})^{-2\gamma}}{(\sum_{i=1}^k (R_i/R_{k+1})^{-\gamma})^2}\sigma^2(X)\right] \\
&\quad + k^2\mathbb{E}\left(a^2(X)\mathbb{E}^2\left[\frac{R_1^2(R_1/R_{k+1})^{-\gamma}}{\sum_{i=1}^k (R_i/R_{k+1})^{-\gamma}}\bigg| X\right]\right) + o.
\end{aligned}
$$

*For classification, under A.1 to A.5, the excess risk w.r.t. $\gamma$ becomes*

$$
Regret(k,n,\gamma) = \frac{1}{4k}B_1\mathbb{E}s^2_{k,n,\gamma}(X) + \int_S \frac{f(x_0)}{\|\dot\eta(x_0)\|}a^2(x_0)t^2_{k,n,\gamma}(x_0)d\mathrm{Vol}^{d-1}(x_0) + o,
$$

*where*

$$
\begin{aligned}
B_1 &= \int_S \frac{f(x_0)}{\|\dot\eta(x_0)\|}d\mathrm{Vol}^{d-1}(x_0), \\
s^2_{k,n,\gamma}(x) &= \frac{\mathbb{E}(R_1/R_{k+1})^{-2\gamma}}{\mathbb{E}^2(R_1/R_{k+1})^{-\gamma}}, \\
t_{k,n,\gamma}(x) &= \frac{\mathbb{E}R_1^2(R_1/R_{k+1})^{-\gamma}}{\mathbb{E}(R_1/R_{k+1})^{-\gamma}}, \\
a(x) &= \frac{1}{f(x)d}\left\{\sum_{j=1}^d [\eta_j(x)f_j(x) + \eta_{j,j}(x)f(x)/2]\right\}.
\end{aligned}
$$

## C.1 REGRESSION

Rewrite the interpolated-NN estimate at $x$ given the distance to the $k+1$th neighbor $R_{k+1}$, interpolation level $\gamma$ as

$$
S_{k,n,\gamma}(x, R_{k+1}) = \sum_{i=1}^k W_i Y_i,
$$

where the weighting scheme is defined as

$$
W_i = \frac{(R_i/R_{k+1})^{-\gamma}}{\sum_{i=1}^k (R_i/R_{k+1})^{-\gamma}}.
$$

For regression, we decompose MSE into bias square and variance, where

$$
\mathbb{E}[(S_{k,n,\gamma}(x, R_{k+1}) - \eta(x))^2|x] = \mathbb{E}\left[\sum_{i=1}^k W_i(\eta(X_i) - \eta(x))\right]^2 + \mathbb{E}\left[\sum_{i=1}^k W_i(Y_i - \eta(X_i))\right]^2,
$$

in which the bias square can be rewritten as

$$
\mathbb{E}\left[\sum_{i=1}^k W_i(\eta(X_i) - \eta(x))\right]^2 = k\mathbb{E}(W_1(\eta(X_1) - \eta(x)))^2 + (k^2 - k)\mathbb{E}^2(W_1(\eta(X_1) - \eta(x))),
$$

and the variance can be approximated as

$$
\mathbb{E}\left[\sum_{i=1}^k W_i(Y_i - \eta(X_i))\right]^2 = k\mathbb{E}W_1^2\sigma(X_1)^2 = k\sigma(x)^2\mathbb{E}W_1^2 + o.
$$

Following a procedure similar as *Step 1* for classification, i.e., use Taylor expansion to approximate the bias square, we obtain that for some function $a$, the bias becomes

$$
\mathbb{E}W_1(\eta(X_1) - \eta(x)) = a(x)\mathbb{E}W_1^2 R_1^2 + o.
$$

As a result, the MSE of interpolated-NN estimate given $x$ becomes,

$$
\mathbb{E}[(S_{k,n,\gamma}(x, R_{k+1}) - \eta(x))^2|x] = k\sigma(x)^2\mathbb{E}W_1^2 + k^2 a(x)^2\mathbb{E}^2 W_1^2 R_1^2 + o.
$$

Finally we integrate MSE over the whole support.

---

[1] The notation "$A + o$" is understood as $A + o = A(1 + o(1))$.

## C.2 CLASSIFICATION

The main structure of the proof follows Samworth et al. (2012). As the whole proof is long, we provide a brief summary in Section C.2.1 to describe things we will do in each step, then in Section C.2.2 we will present the details in each step.

### C.2.1 BRIEF SUMMARY

*Step 1*: denote i.i.d random variables $Z_i(x, R_{k+1})$ for $i = 1, \ldots, k$ where

$$Z_i(x, R_{k+1}) = \frac{(R_i/R_{k+1})^{-\gamma}(Y(X_i) - 1/2)}{\mathbb{E}(R_i(x)/R_{k+1}(x))^{\gamma}},$$

then the probability of classifying as 0 becomes

$$P(S_{k,n,\gamma}(x) < 1/2) = P\left(\sum_{i=1}^{k} Z_i(x, R_{k+1}) < 0\right).$$

The mean and variance of $Z_i(x, R_{k+1})$ can be obtained through Taylor expansion of $\eta$ and density function of $x$:

$$
\begin{aligned}
\mathbb{E}(Z_1(x, R_{k+1})) &= \eta(x) + a(x)\frac{\mathbb{E}R_1^2(R_1/R_{k+1})^{-\gamma}}{\mathbb{E}(R_1/R_{k+1})^{-\gamma}} + o \\
Var(Z_1(x, R_{k+1})) &= \frac{1}{4}\frac{\mathbb{E}(R_1/R_{k+1})^{-2\gamma}}{\mathbb{E}^2(R_1/R_{k+1})^{-\gamma}} + o,
\end{aligned}
$$

for some function $a$. The smoothness conditions are assumed in A.4 and A.5.

Note that on the denominator of $Z_i$, there is an expectation $\mathbb{E}(R_i(x)/R_{k+1}(x))^{\gamma}$. From later calculation in Corollary 1, the value of this expectation in fact has little changes given or without a condition of $R_{k+1}$, and it is little affected by $x$ either.

*Step 2*: One can rewrite Regret as

$$\int_{\mathbb{R}^d}\left(P\left(\sum_{i=1}^{k} W_i Y_i \leq \frac{1}{2}\right) - 1_{\{\eta(x)<1/2\}}\right) d\bar{P}(x).$$

From Assumption A.2, A.4, the region where $\hat{\eta}$ is likely to make a wrong prediction is near $\{x|\eta(x) = 1/2\}$, thus we use tube theory to transform the integral of Regret over the $d$-dimensional space into a tube, i.e.,

$$
\begin{aligned}
&\int_{\mathbb{R}^d}\left(P\left(\sum_{i=1}^{k} W_i Y_i \leq \frac{1}{2}\right) - 1\{\eta(x) < 1/2\}\right) d\bar{P}(x) \\
=\ &\{1 + o(1)\}\int_{\mathcal{S}}\int_{-\epsilon}^{\epsilon} t\|\dot{\Psi}(x_0)\|\left(P(S_{k,n}(x_0^t) < 1/2) - 1_{\{t<0\}}\right) dt d\text{Vol}^{d-1}(x_0) + o.
\end{aligned}
$$

The term $\epsilon$ will be defined in detail in appendix. Basically, when $\epsilon$ is within a suitable range, the integral over $t$ will not depend on $\epsilon$ asymptotically.

*Step 3*: given $R_{k+1}$ and $x$, the nearest $k$ neighbors are i.i.d. random variables distributed in $B(x, R_{k+1})$, thus we use non-uniform Berry-Esseen Theorem to get the Gaussian approximation of the probability of wrong prediction:

$$
\begin{aligned}
&\int_{\mathcal{S}}\int_{-\epsilon}^{\epsilon} t\|\dot{\Psi}(x_0)\|\left(P(S_{k,n}(x_0^t) < 1/2) - 1_{\{t<0\}}\right) dt d\text{Vol}^{d-1}(x_0) \\
=\ &\int_{\mathcal{S}}\int_{-\epsilon}^{\epsilon} t\|\dot{\Psi}(x_0)\|\mathbb{E}_{R_{k+1}}\left(\Phi\left(\frac{-k\mathbb{E}Z_1(x_0^t, R_{k+1})}{\sqrt{kVar(Z_1(x_0^t, R_{k+1}))}}\right) - 1_{\{t<0\}}\right) dt d\text{Vol}^{d-1}(x_0) + o.
\end{aligned}
$$

*Step 4*: take expectation over all $R_{k+1}$, and integral the Gaussian probability over the tube to obtain

$$\int_{\mathcal{S}}\int_{-\epsilon}^{\epsilon} t\|\dot{\Psi}(x_0)\|\mathbb{E}_{R_{k+1}}\left(\Phi\left(\frac{-k\mathbb{E}Z_1(x_0^t,R_{k+1})}{\sqrt{kVar(Z_1(x_0^t,R_{k+1}))}}\right) - 1_{\{t<0\}}\right)dtd\text{Vol}^{d-1}(x_0)$$

$$= \int_{\mathcal{S}}\int_{\mathbb{R}} t\|\dot{\Psi}(x_0)\|\left(\Phi\left(-\frac{t\|\dot{\eta}(x_0)\|}{\sqrt{s_{k,n,\gamma}^2/k}} - \frac{\mathbb{E}(R_1/R_{k+1})^{-\gamma}a(x_0^t)R_1^2}{\sqrt{s_{k,n,\gamma}^2/k}}\right) - 1_{\{t<0\}}\right)dtd\text{Vol}^{d-1}(x_0) + o$$

$$= \frac{B_1}{4k}\frac{\mathbb{E}(R_1/R_{k+1})^{-2\gamma}}{\mathbb{E}^2(R_1/R_{k+1})^{-\gamma}} + \int_S \frac{\|\dot{\Phi}(x_0)\|}{\|\dot{\eta}(x_0)\|^2}a^2(x_0)\frac{\mathbb{E}^2(R_1/R_{k+1})^{-\gamma}R_1^2}{\mathbb{E}^2(R_1/R_{k+1})^{-\gamma}}d\text{Vol}^{d-1}(x_0) + o$$

$$= \frac{1}{4k}B_1\mathbb{E}s_{k,n,\gamma}^2 + \int_S \frac{\|\dot{\Phi}(x_0)\|}{\|\dot{\eta}(x_0)\|^2}a^2(x_0)t_{k,n,\gamma}^2 d\text{Vol}^{d-1}(x_0) + o.$$

### C.2.2 DETAILS

Denote $a_d$ is the Euclidean ball volume parameter

$$a_d = \text{Vol}(B(0,1)) = (\pi/2)^{d/2}/\Gamma(d/2+1).$$

Define $p = k/n$ and $r_{2p} = \sup_x \mathbb{E}R_{2k}(x)$. Denote $E$ be the set that there exists $R_i$ such that $R_i > r_{2p}$, then for some constant $c > 0$,

$$r_{2p} = \frac{c}{a_d^{1/d}c_0^{1/d}}\left(\frac{2k}{n}\right)^{1/d}.$$

Hence from Claim A.5 in Belkin et al. (2018), there exist $c_1$ and $c_2$ satisfying

$$P(E) \le c_1 k\exp(-c_2 k).$$

*Step 1:* in this step, we figure out the i.i.d. random variable in our problem, and calculate its mean and variance given $x$.

Denote

$$Z_i(x,R_{k+1}) = \frac{(R_i/R_{k+1})^{-\gamma}(Y(X_i)-1/2)}{\mathbb{E}(R_i/R_{k+1})^{\gamma}}, \tag{2}$$

then the dominant part we want to integrate becomes

$$P\left(S_{k,n}(x,R_{k+1}) \le \frac{1}{2}\right)$$

$$= P\left(\sum_{i=1}^{k}(R_i/R_{k+1})^{-\gamma}(Y(X_i)-1/2) < 0 \Big| R_{k+1}\right)$$

$$= P\left(\frac{\sum_{i=1}^{k}Z_i(x,R_{k+1}) - k\mathbb{E}Z_1(x,R_{k+1})}{\sqrt{kVar(Z_1(x,R_{k+1}))}} < \frac{-k\mathbb{E}Z_1(x,R_{k+1})}{\sqrt{kVar(Z_1(x,R_{k+1}))}}\Big| R_{k+1}\right).$$

Therefore, one can adopt non-uniform Berry-Essen Theorem to approximate the probability using normal distribution. Unlike Samworth et al. (2012) in which $\mathbb{E}Y(X_i)$ is calculated, since the i.i.d. item in non-uniform Berry-Essen Theorem is $Z$ rather than $Y$, we now calculate mean and variance of $Z$. Under $R_{k+1}$,

$$\mu_{k,n,\gamma}(x,R_{k+1}) := \mathbb{E}Z_1(x,R_{k+1}) = \frac{\mathbb{E}(R_1/R_{k+1})^{-\gamma}(Y(X_1)-1/2)}{\mathbb{E}(R_1/R_{k+1})^{-\gamma}}$$

$$= \frac{\mathbb{E}(R_1/R_{k+1})^{-\gamma}(\eta(X_1)-1/2)}{\mathbb{E}(R_1/R_{k+1})^{-\gamma}},$$

and

$$\mathbb{E}Z_1^2(x, R_{k+1}) = \frac{\mathbb{E}(R_1/R_{k+1})^{-2\gamma}(Y(X_1) - 1/2)^2}{\mathbb{E}^2(R_1/R_{k+1})^{-\gamma}}$$

$$= \frac{\mathbb{E}(R_1/R_{k+1})^{-2\gamma}}{4\mathbb{E}^2(R_i/R_{k+1})^{-\gamma}},$$

$$\sigma_{k,n,\gamma}^2(x, R_{k+1}) := Var(Z_1(x, R_{k+1})).$$

Then the mean and variance of $Z_i$ can be calculated as

$$\mu_{k,n}(x_0^t, R_{k+1}) = \mathbb{E}Z_1(x_0^t, R_{k+1}) + \frac{1}{2} = \frac{\mathbb{E}(R_1/R_{k+1})^{-\gamma}\eta(X_1)}{\mathbb{E}(R_1/R_{k+1})^{-\gamma}} + \frac{1}{2}$$

$$= \frac{\mathbb{E}(R_1/R_{k+1})^{-\gamma}\left(\eta(x_0^t) + (X_1 - x_0^t)^\top \dot{\eta}(x_0^t) + 1/2(X_1 - x_0^t)^\top \ddot{\eta}(x_0^t)(X_1 - x_0^t)\right)}{\mathbb{E}(R_1/R_{k+1})^{-\gamma}} + o(R_{k+1}^3)$$

$$= \eta(x_0^t) + \frac{\mathbb{E}(R_1/R_{k+1})^{-\gamma}(X_1 - x_0^t)^\top \dot{\eta}(x_0^t)}{\mathbb{E}(R_1/R_{k+1})^{-\gamma}}$$

$$+ \frac{1}{2}\frac{\mathbb{E}(R_1/R_{k+1})^{-\gamma}tr[\ddot{\eta}(x_0^t)\left((X_1 - x_0^t)(X_1 - x_0^t)^\top\right)]}{\mathbb{E}(R_1/R_{k+1})^{-\gamma}} + O(R_{k+1}^3).$$

Fixing $R_1$ and $R_{k+1}$, we have

$$\mathbb{E}((X_1 - x_0^t)^\top \dot{\eta}(x_0^t)|R_1)$$

$$= \int (x - x_0^t)^\top \dot{\eta}(x_0^t) f(x|x_0^t, R_{k+1}) dx$$

$$= \int (x - x_0^t)^\top \dot{\eta}(x_0^t)[f(x_0^t|x_0^t, R_1) + f'(x_0|x_0^t, R_1)^\top (x - x_0^t) + o] dx$$

$$= 0 + \int (x - x_0^t)^\top \dot{\eta}(x_0^t) f'(x_0^t|x_0^t, R_1)^\top (x - x_0^t) dx + o$$

$$= tr\left(\dot{\eta}(x_0^t) f'(x_0^t|x_0^t, R_1)^\top \int (x - x_0^t)(x - x_0^t)^\top dx\right) + o \tag{3}$$

and

$$tr\left(\frac{1}{2}\ddot{\eta}(x_0^t)\mathbb{E}\left((X_1 - x_0^t)(X_1 - x_0^t)^\top|R_1\right)\right)$$

$$= tr\left(\frac{1}{2}\ddot{\eta}(x_0^t)\int (x - x_0^t)(x - x_0^t)^\top f(x|x_0^t, R_1) dx\right)$$

$$= tr\left(\frac{1}{2}\ddot{\eta}(x_0^t)\int (x - x_0^t)(x - x_0^t)^\top [f(x_0^t|x_0^t, R_1) + f'(x_0|x_0^t, R_1)^\top (x - x_0^t) + o] dx\right)$$

$$= tr\left(\frac{f(x_0^t|x_0^t, R_1)}{2}\ddot{\eta}(x_0^t)\int (x - x_0^t)(x - x_0^t)^\top dx\right) + o, \tag{4}$$

Then taking function $a(x)$ for $x$ such that

$$a(x_0^t)R_1^2 = \mathbb{E}((X_1 - x_0^t)^\top \dot{\eta}(x_0^t)|R_1) + tr\left(\frac{1}{2}\ddot{\eta}(x_0^t)\mathbb{E}\left((X_i - x_0^t)(X_i - x_0^t)^\top|R_1\right)\right) + o,$$

which can be satisfied through taking

$$a(x) = \frac{1}{f(x)^{1+2/d}d}\left\{\sum_{j=1}^d [\dot{\eta}_j(x)\dot{f}_j(x) + \ddot{\eta}_{j,j}(x)f(x)/2]\right\}.$$

The difference caused by the value of $R_1$ is only a small order term.

Finally,

$$\mu_{k,n,\gamma}(x_0^t, R_{k+1}) = \eta(x_0) + t\|\dot\eta(x_0)\| + a(x_0)t_{k,n,\gamma}(R_{k+1}) + o.$$

*Step 2:* in this step we construct a tube based on the set $\mathcal{S} = \{x|\eta(x) = 1/2\}$, then figure out that the part of Regret outside this tube is a remainder term.

Assume $\epsilon_{k,n}$ satisfies $s_{k,n,\gamma} = o(\epsilon_{k,n})$ and $\epsilon_{k,n} = o(s_{k,n,\gamma}k^{1/2})$, then the residual terms throughout the following steps will be $o(s_{k,n,\gamma}^2 + t_{k,n,\gamma}^2)$. Hence although the choice of $\epsilon_{k,n}$ is different among choices of $k$ and $n$, this does not affect the rate of Regret. Note that we ignore the arguments $x$ and $R_{k+1}$ as from A.2 and A.4, $s_{k,n,\gamma}^2(x, R_{k+1}) \asymp 1/k$ and $t_{k,,n,\gamma}(x, R_{k+1}) \asymp (k/n)^{2/d}$ for all $x$ while $R_{k+1} \asymp (k,n)^{2/d}$ in probability.

By Samworth et al. (2012), recall that $\Psi(x) = d(\pi_1 P_1(x) - \pi_2 P_2(x))$, then

$$\int_{\mathbb{R}^d} \left( P\left(\sum_{i=1}^k W_i Y_i \le \frac{1}{2}\right) - 1_{\{\eta(x)<1/2\}} \right) d\bar{P}(x)$$

$$= \{1 + o(1)\} \int_{\mathcal{S}} \int_{-\epsilon_{k,n}}^{\epsilon_{k,n}} t\|\dot\Psi(x_0)\| \left( P(S_{k,n}(x_0^t) < 1/2) - 1_{\{t<0\}} \right) dt d\mathrm{Vol}^{d-1}(x_0) + r_1,$$

where

$$r_1 = \int_{\mathbb{R}^d \setminus \mathcal{S}^{\epsilon_{k,n}}} \left( P\left(\sum_{i=1}^k W_i Y_i \le \frac{1}{2}\right) - 1_{\{\eta(x)<1/2\}} \right) d\bar{P}(x)$$

$$= \int_{\mathbb{R}^d \setminus \mathcal{S}^{\epsilon_{k,n}}} \mathbb{E}_{R_{k+1}} \left( P\left(\sum_{i=1}^k Z_i(x, R_{k+1}) < 0\right) - 1_{\{\eta(x)<1/2\}} \right) d\bar{P}(x).$$

For $r_1$,

$$0 \ge \int_{\mathbb{R}^d \setminus \mathcal{S}^{\epsilon_{k,n}} \cap \{x|\eta(x)<1/2\}} \mathbb{E}_{R_{k+1}} \left( P\left(\sum_{i=1}^k Z_i(x, R_{k+1}) \le 0\right) - 1_{\{\eta(x)<1/2\}} \right) d\bar{P}(x)$$

$$= -\int_{\mathbb{R}^d \setminus \mathcal{S}^{\epsilon_{k,n}} \cap \{x|\eta(x)<1/2\}} \mathbb{E}_{R_{k+1}} \left( P\left(\sum_{i=1}^k Z_i(x, R_{k+1}) - k\mathbb{E}Z_1(x, R_{k+1}) > -k\mathbb{E}Z_1(x, R_{k+1})\right) \right) d\bar{P}(x).$$

Using non-uniform Berry-Essen Theorem, when $\mathbb{E}Z_1^3(x, R_{k+1}) < \infty$, i.e. $\gamma < d/3$, it becomes

$$-\int_{\mathbb{R}^d \setminus \mathcal{S}^{\epsilon_{k,n}} \cap \{x|\eta(x)<1/2\}} \mathbb{E}_{R_{k+1}} \left( P\left(\sum_{i=1}^k Z_i(x, R_{k+1}) - k\mathbb{E}Z_1(x, R_{k+1}) > -k\mathbb{E}Z_1(x, R_{k+1})\right) \right) d\bar{P}(x)$$

$$\le \int_{\mathbb{R}^d \setminus \mathcal{S}^{\epsilon_{k,n}} \cap \{x|\eta(x)<1/2\}} \mathbb{E}_{R_{k+1}} \bar{\Phi}\left( -\frac{\sqrt{k}\mathbb{E}Z_1(x, R_{k+1})}{Var(Z_1(x, R_{k+1}))} \right) d\bar{P}(x)$$

$$+ c_1 \frac{1}{\sqrt{k}} \int_{\mathbb{R}^d \setminus \mathcal{S}^{\epsilon_{k,n}} \cap \{x|\eta(x)<1/2\}} \mathbb{E}_{R_{k+1}} \frac{1}{1 + k^{3/2}|\mathbb{E}Z_1(x, R_{k+1})|^3} d\bar{P}(x),$$

where $\bar{\Phi}(x) = 1 - \Phi(x)$. Since $s_{k,n,\gamma}(x, R_{k+1}) = o(\epsilon_{k,n})$, $r_1 = o(s_{k,n,\gamma}^2(x, R_{k+1}))$.

By the definition of $\epsilon_{k,n}$, we have

$$\exp(-\epsilon_{k,n}^2/s_{k,n,\gamma}^2(x, R_{k+1})) = o(s_{k,n,\gamma}^2) + o(1/k),$$

$$\inf_{x \in \mathbb{R}^d \setminus \mathcal{S}^{\epsilon_{k,n}}} |\eta(x) - 1/2| \ge c_3 \epsilon_{k,n}.$$

As a result, using Berstain inequality, $r_1$ is a smaller order term compared with $s_{k,n,\gamma}^2$ when $s_{k,n,\gamma}^2(R_{k+1}) = o(1)$, hence $r_1 = o(s_{k,n,\gamma}^2)$.

*Step 3*: now we apply non-uniform Berry-Esseen Theorem. From *Step 3*, we have

$$\int_{\mathcal{S}} \int_{-\epsilon_{k,n}}^{\epsilon_{k,n}} \mathbb{E}_{R_{k+1}} t \|\dot{\Psi}(x_0)\| \left( P(S_{k,n}(x_0^t) < 1/2 | R_{k+1}) - 1_{\{t<0\}} \right) dt d\text{Vol}^{d-1}(x_0)$$

$$= \int_{\mathcal{S}} \int_{-\epsilon_{k,n}}^{\epsilon_{k,n}} \mathbb{E}_{R_{k+1}} t \|\dot{\Psi}(x_0)\| \left( \Phi \left( \frac{-k \mathbb{E} Z_1(x_0^t, R_{k+1})}{\sqrt{k Var(Z_1(x_0^t, R_{k+1}))}} \right) - 1_{\{t<0\}} \right) dt d\text{Vol}^{d-1}(x_0) + r_2,$$

where based on non-uniform Berry-Esseen Theorem:

$$\left| P \left( \frac{\sum_{i=1}^k Z_i(x_0^t, R_{k+1}) - k \mathbb{E} Z_1(x_0^t, R_{k+1})}{\sqrt{k Var(Z_1(x_0^t, R_{k+1}))}} < \frac{-k \mathbb{E} Z_1(x_0^t, R_{k+1})}{\sqrt{k Var(Z_1(x_0^t, R_{k+1}))}} \right) - \Phi \left( \frac{-k \mathbb{E} Z_1(x_0^t, R_{k+1})}{\sqrt{k Var(Z_1(x_0^t, R_{k+1}))}} \right) \right|$$

$$\leq c \frac{k \mathbb{E} |Z_1(x_0^t, R_{k+1})|^3}{k^{3/2} Var^{3/2}(Z_1(x_0^t, R_{k+1}))} \frac{1}{1 + \left| \frac{-k \mathbb{E} Z_1(x_0^t, R_{k+1})}{\sqrt{k Var(Z_1(x_0^t, R_{k+1}))}} \right|^3},$$

and

$$r_2 \leq \int_{\mathcal{S}} \int_{-\epsilon_{k,n}}^{\epsilon_{k,n}} \mathbb{E}_{R_{k+1}} t \|\dot{\Psi}(x_0)\| \frac{k \mathbb{E} |Z_1(x_0^t, R_{k+1})|^3}{k^{3/2} Var^{3/2}(Z_1(x_0^t, R_{k+1}))} \frac{1}{1 + \left| \frac{-k \mathbb{E} Z_1(x_0^t, R_{k+1})}{\sqrt{k Var(Z_1(x_0^t, R_{k+1}))}} \right|^3} dt d\text{Vol}^{d-1}(x_0).$$

For $r_2$,

$$r_2 \leq \int_{\mathcal{S}} \int_{-\epsilon_{k,n}}^{\epsilon_{k,n}} \mathbb{E}_{R_{k+1}} t \|\dot{\Psi}(x_0)\| \frac{k \mathbb{E} |Z_1(x_0^t, R_{k+1})|^3}{k^{3/2} Var^{3/2}(Z_1(x_0^t, R_{k+1}))} \frac{1}{1 + \left| \frac{-k \mathbb{E} Z_1(x_0^t, R_{k+1})}{\sqrt{k Var(Z_1(x_0^t, R_{k+1}))}} \right|^3} dt d\text{Vol}^{d-1}(x_0)$$

$$= \frac{c_1}{\sqrt{k}} \int_{\mathcal{S}} \int_{-\epsilon_{k,n}}^{\epsilon_{k,n}} \mathbb{E}_{R_{k+1}} t \|\dot{\Psi}(x_0)\| \frac{1}{1 + \left| \frac{-k \mathbb{E} Z_1(x_0^t, R_{k+1})}{\sqrt{k Var(Z_1(x_0^t, R_{k+1}))}} \right|^3} dt d\text{Vol}^{d-1}(x_0)$$

$$= \frac{c_2}{\sqrt{k}} \int_{\mathcal{S}} \int_{|t| < s_{k,n,\gamma}(x)} t \|\dot{\Psi}(x_0)\| dt d\text{Vol}^{d-1}(x_0)$$

$$+ \frac{c_3}{\sqrt{k}} \int_{\mathcal{S}} \int_{s_{k,n,\gamma}(x) < |t| < \epsilon_{k,n}} t \|\dot{\Psi}(x_0)\| \frac{1}{1 + k^{3/2} t^3} dt d\text{Vol}^{d-1}(x_0)$$

$$= o(s_{k,n,\gamma}^2).$$

*Step 4*: the integral becomes

$$\int_{\mathcal{S}} \mathbb{E}_{R_{k+1}} \int_{-\epsilon_{k,n}}^{\epsilon_{k,n}} t \|\dot{\Psi}(x_0)\| \left( \Phi \left( \frac{-k \mathbb{E} Z_1(x_0^t, R_{k+1})}{\sqrt{k Var(Z_1(x_0^t, R_{k+1}))}} \right) - 1_{\{t<0\}} \right) dt d\text{Vol}^{d-1}(x_0)$$

$$= \int_{\mathcal{S}} \mathbb{E}_{R_{k+1}} \int_{-\epsilon_{k,n}}^{\epsilon_{k,n}} t \|\dot{\Psi}(x_0)\| \left( \Phi \left( -\frac{t \|\dot{\eta}(x_0)\|}{\sqrt{s_{k,n,\gamma}^2(x, R_{k+1})/k}} - \frac{\mathbb{E}(R_1/R_{k+1})^{-\gamma} a(x_0^t) R_1^2}{\sqrt{s_{k,n,\gamma}^2(x, R_{k+1})/k}} \right) - 1_{\{t<0\}} \right) dt d\text{Vol}^{d-1}(x_0)$$

$$+ r_3 + o$$

$$= \int_{\mathcal{S}} \mathbb{E}_{R_{k+1}} \int_{\mathbb{R}} t \|\dot{\Psi}(x_0)\| \left( \Phi \left( -\frac{t \|\dot{\eta}(x_0)\|}{\sqrt{s_{k,n,\gamma}^2(x, R_{k+1})/k}} - \frac{\mathbb{E}(R_1/R_{k+1})^{-\gamma} a(x_0^t) R_1^2}{\sqrt{s_{k,n,\gamma}^2(x, R_{k+1})/k}} \right) - 1_{\{t<0\}} \right) dt d\text{Vol}^{d-1}(x_0)$$

$$+ r_3 + r_4 + o$$

$$= \int_{\mathcal{S}} \int_{\mathbb{R}} t \|\dot{\Psi}(x_0)\| \left( \Phi \left( -\frac{t \|\dot{\eta}(x_0)\|}{\sqrt{s_{k,n,\gamma}^2(x)/k}} - \frac{\mathbb{E}(R_1/R_{k+1})^{-\gamma} a(x_0^t) R_1^2}{\sqrt{s_{k,n,\gamma}^2(x)/k}} \right) - 1_{\{t<0\}} \right) dt d\text{Vol}^{d-1}(x_0)$$

$$+ r_3 + r_4 + r_5 + o$$

$$= \frac{B_1}{4k} \frac{\mathbb{E}(R_1/R_{k+1})^{-2\gamma}}{\mathbb{E}^2(R_1/R_{k+1})^{-\gamma}} + \int_{\mathcal{S}} \frac{f(x_0)}{\|\dot{\eta}(x_0)\|} a^2(x_0) \frac{\mathbb{E}^2(R_1/R_{k+1})^{-\gamma} R_1^2}{\mathbb{E}^2(R_1/R_{k+1})^{-\gamma}} d\text{Vol}^{d-1}(x_0) + r_3 + r_4 + r_5 + o.$$

Note that $\|\dot{\Psi}(x_0)\|/\|\dot{\eta}(x_0)\| = 2f(x_0)$. The last step follows Proposition 4 and the fact that $R_{k+1}$ does not affect the dominant parts. The term $\mathbb{E}((R_1/R_{k+1})^{-\gamma}|R_{k+1})$ is almost the same for all $R_{k+1}$. For the small order terms, following Samworth et al. (2012) we obtain

$$
\begin{aligned}
r_3 &= \int_{\mathcal{S}} \mathbb{E}_{R_{k+1}} \int_{-\epsilon_{k,n}}^{\epsilon_{k,n}} t\|\dot{\Psi}(x_0)\| \Bigg( \Phi\left( \frac{-k\mathbb{E}Z_1(x_0^t, R_{k+1})}{\sqrt{kVar(Z_1(x_0^t, R_{k+1}))}} \right) \\
&\qquad\qquad -\Phi\left( -\frac{t\|\dot{\eta}(x_0)\|}{\sqrt{s_{k,n,\gamma}^2(x, R_{k+1})/k}} - \frac{\mathbb{E}(R_1/R_{k+1})^{-\gamma}a(x_0^t)R_1^2}{\sqrt{s_{k,n,\gamma}^2(x, R_{k+1})/k}} \right) \Bigg) dt d\text{Vol}^{d-1}(x_0), \\
&= o(s_{k,n,\gamma}^2 + t_{k,n,\gamma}^2),
\end{aligned}
$$

and

$$
\begin{aligned}
r_4 &= \int_{\mathcal{S}} \frac{\|\dot{\Psi}(x_0)\|}{\|\dot{\eta}(x_0)\|^2} \mathbb{E}_{R_{k+1}} \int_{\mathbb{R}\backslash[-\epsilon_{k,n},\epsilon_{k,n}]} \\
&\qquad t\|\dot{\Psi}(x_0)\| \left( \Phi\left( -\frac{t\|\dot{\eta}(x_0)\|}{\sqrt{s_{k,n,\gamma}^2/k}} - \frac{\mathbb{E}(R_1/R_{k+1})^{-\gamma}a(x_0^t)R_1^2}{\sqrt{s_{k,n,\gamma}^2/k}} \right) - 1_{\{v<0\}} \right) dt d\text{Vol}^{d-1}(x_0) \\
&= o(s_{k,n,\gamma}^2).
\end{aligned}
$$

The term $r_5$ is the difference between the normal probability given $R_{k+1}$ and the one after taking expectation. Similar with Cannings et al. (2017), for each $x$, when $|t| < \epsilon_{k,n}$, we have

$$
\begin{aligned}
&\mathbb{E}_{R_{k+1}} \Phi\left( -\frac{t\|\dot{\eta}(x_0)\|}{\sqrt{s_{k,n,\gamma}^2(x, R_{k+1})/k}} - \frac{\mathbb{E}(R_1/R_{k+1})^{-\gamma}a(x_0^t)R_1^2}{\sqrt{s_{k,n,\gamma}^2(x, R_{k+1})/k}} \right) \\
&= \Phi\left( -\frac{t\|\dot{\eta}(x_0)\|}{\sqrt{s_{k,n,\gamma}^2(x, R_{k+1})/k}} - \frac{\mathbb{E}(R_1/R_{k+1})^{-\gamma}a(x_0^t)R_1^2}{\sqrt{s_{k,n,\gamma}^2(x, R_{k+1})/k}} \right) + O\left( kVar(t_{k,n,\gamma}(x, R_{k+1})) \right) + o.
\end{aligned}
$$

Following step 3 in Cannings et al. (2017), we obtain

$$
Var(t_{k,n,\gamma}(x, R_{k+1})) \le \frac{1}{k^2} \sum_{j=1}^{k} \mathbb{E}(\eta(X_1) - \eta(x))^2 = O\left( \frac{1}{k} r_{2p}^2 \right).
$$

For the case when $|t| \gg s_{k,n,\gamma} + t_{k,n,\gamma}$, differentiate normal cdf twice still leads to very small probability, thus for each $x$, we have

$$
\begin{aligned}
&\int t\mathbb{E}_{R_{k+1}} \Phi\left( -\frac{t\|\dot{\eta}(x_0)\|}{\sqrt{s_{k,n,\gamma}^2(x, R_{k+1})/k}} - \frac{\mathbb{E}(R_1/R_{k+1})^{-\gamma}a(x_0^t)R_1^2}{\sqrt{s_{k,n,\gamma}^2(x, R_{k+1})/k}} \right) dt \\
&= \int t\Phi\left( -\frac{t\|\dot{\eta}(x_0)\|}{\sqrt{s_{k,n,\gamma}^2(x, R_{k+1})/k}} - \frac{\mathbb{E}(R_1/R_{k+1})^{-\gamma}a(x_0^t)R_1^2}{\sqrt{s_{k,n,\gamma}^2(x, R_{k+1})/k}} \right) dt + o(s_{k,n,\gamma}^2 + t_{k,n,\gamma}^2).
\end{aligned}
$$

## D   PROOF OF THEOREM 1

To show the ratio $\text{MSE}(\gamma, n)/\text{MSE}(0, n)$ and $\text{Regret}(\gamma, n)/\text{Regret}(0, n)$ asymptotically converges to a constant, we need to figure out the corresponding optimum $k$ for each scenario.

For classification, given $x$, we know that if $X$ follows multi-dimensional uniform distribution with density $1/f(x)$, for some constant $c_d$ that only depends on $d$,

$$\mathbb{E}(R_1/R_{k+1})^{-2\gamma} = \mathbb{E}_{R_{k+1}}d\int_0^{R_{k+1}}\left(\frac{r}{R_{k+1}}\right)^{-2\gamma}r^{d-1}dr + o = \frac{d}{d-2\gamma} + o,$$

$$\mathbb{E}(R_1/R_{k+1})^{-\gamma} = \frac{d}{d-\gamma} + o,$$

$$\mathbb{E}(R_1/R_{k+1})^{-\gamma}R_1^2 = \mathbb{E}(R_1/R_{k+1})^{2-\gamma}R_{k+1}^2 = c_d\left(\frac{k}{nf(x)}\right)^{\frac{2}{d}}\frac{d}{d+2-\gamma} + o.$$

Since $R_i \to 0$, this approximation still holds if $X$ follow other distributions. As a result, through figuring out the optimum $k$ for 0 and $\gamma$, we have

$$\frac{\text{Regret}(n,\gamma)}{\text{Regret}(n,0)} \to \left(1 - \frac{\gamma^2}{d(d-2\gamma)}\right)^{\frac{4}{d+4}}\left(\frac{(d-\gamma)^2}{(d+2-\gamma)^2}\frac{(d+2)^2}{d^2}\right)^{\frac{d}{d+4}}.$$

For regression, one more step needed compared with classification is to evaluate

$$k\mathbb{E}\left[\frac{(R_1/R_{k+1})^{-2\gamma}}{(\sum_{i=1}^k(R_i/R_{k+1})^{-\gamma})^2}\right]\mathbb{E}\sigma(X)^2 + k^2\mathbb{E}\left(a^2(X)\mathbb{E}^2\left[\frac{R_1^2(R_1/R_{k+1})^{-\gamma}}{\sum_{i=1}^k(R_i/R_{k+1})^{-\gamma}}\right]\right).$$

The sum of ratios $\sum_{i=1}^k(R_i/R_{k+1})^{-\gamma}$ is hard to evaluated directly in the denominator, hence we use upper bound and lower bound on it. Since $d - 3\gamma > 0$, using non-uniform Berry-Essen Theorem, given $R_{k+1}$, we have

$$\left|P\left(\sum_{i=1}^k(R_i/R_{k+1})^{-\gamma} - k\mathbb{E}(R_1/R_{k+1})^{-\gamma} > \xi\right) - \bar{\Phi}\left(\frac{\xi}{\sqrt{kVar((R_1/R_{k+1})^{-\gamma})}}\Big|R_{k+1}\right)\right| \leq \frac{c}{1+(\xi/\sqrt{k})^3}.$$

Therefore, taking $\xi = \delta_k k\mathbb{E}(R_1/R_{k+1})^{-\gamma}$,

$$P\left(\sum_{i=1}^k(R_i/R_{k+1})^{-\gamma} > (\delta_k+1)k\mathbb{E}(R_1/R_{k+1})^{-\gamma}\Big|R_{k+1}\right)$$

$$\leq \bar{\Phi}\left(\frac{\delta_k\sqrt{k}\mathbb{E}(R_1/R_{k+1})^{-\gamma}}{\sqrt{Var((R_1/R_{k+1})^{-\gamma})}}\Big|R_{k+1}\right) + \frac{c}{1+(\sqrt{k}\delta_k)^3}.$$

Note that $(R_1/R_{k+1})^{-\gamma}$ is always larger than 1, hence

$$\mathbb{E}\left[\frac{(R_1/R_{k+1})^{-2\gamma}}{(\sum_{i=1}^k(R_i/R_{k+1})^{-\gamma})^2}\right]$$

$$\leq \mathbb{E}_{R_{k+1}}\left[\frac{\mathbb{E}(R_1/R_{k+1})^{-2\gamma}}{(1-\delta_k)^2k^2\mathbb{E}^2(R_1/R_{k+1})^{-\gamma}}\right]$$

$$+\mathbb{E}_{R_{k+1}}\left[P\left(\sum_{i=1}^k(R_i/R_{k+1})^{-\gamma} < (1-\delta_k)k\mathbb{E}(R_1/R_{k+1})^{-\gamma}\Big|R_{k+1}\right)\frac{\mathbb{E}(R_1/R_{k+1})^{-2\gamma}}{k^2}\right] + o$$

$$\leq \frac{1}{k^2(1-\delta_k)^2}\frac{\mathbb{E}(R_1/R_{k+1})^{-2\gamma}}{\mathbb{E}^2(R_1/R_{k+1})^{-\gamma}} + \frac{\mathbb{E}(R_1/R_{k+1})^{-2\gamma}}{k^2}\bar{\Phi}\left(\frac{\delta_k\sqrt{k}\mathbb{E}(R_1/R_{k+1})^{-\gamma}}{\sqrt{Var((R_1/R_{k+1})^{-\gamma})}}\right)$$

$$+\frac{\mathbb{E}(R_1/R_{k+1})^{-2\gamma}}{k^2}\frac{c}{1+(\sqrt{k}\delta_k)^3} + o,$$

while

$$\mathbb{E}\left[\frac{(R_1/R_{k+1})^{-2\gamma}}{(\sum_{i=1}^k(R_i/R_{k+1})^{-\gamma})^2}\right] \geq \frac{1}{k^2(1+\delta_k)^2}\frac{\mathbb{E}(R_1/R_{k+1})^{-2\gamma}}{\mathbb{E}^2(R_1/R_{k+1})^{-\gamma}} - \frac{\mathbb{E}(R_1/R_{k+1})^{-2\gamma}}{k^2}\frac{c}{1+(\sqrt{k}\delta_k)^3} + o.$$

Hence taking $\delta_k$ such that $\delta_k \to 0$ while $\delta_k\sqrt{k} \to \infty$, we have

$$\mathbb{E}\left[\frac{(R_1/R_{k+1})^{-2\gamma}}{(\sum_{i=1}^k (R_i/R_{k+1})^{-\gamma})^2}\right] = \frac{1}{k^2}\frac{(d-\gamma)^2}{d(d-2\gamma)} + o,$$

and similarly

$$\mathbb{E}\left[\frac{R_1^2(R_1/R_{k+1})^{-\gamma}}{\sum_{i=1}^k (R_i/R_{k+1})^{-\gamma}}\right] = \frac{1}{k}\left(\frac{k}{nf(x)}\right)^{\frac{2}{d}}\frac{d-\gamma}{d+2-\gamma} + o.$$

After figuring out the optimum $k$ for $k$NN and interpolated-NN, we finally obtain

$$\frac{\text{MSE}(n,\gamma)}{\text{MSE}(n,0)} \to \left(1+\frac{\gamma^2}{d(d-2\gamma)}\right)^{\frac{4}{d+4}}\left(\frac{(d-\gamma)^2}{(d+2-\gamma)^2}\frac{(d+2)^2}{d^2}\right)^{\frac{d}{d+4}}.$$

## E  PROOF OF THEOREM 2

The proof is similar with Theorem 1 in Sun et al. (2016).

From the definition of CIS, we have

$$
\begin{aligned}
CIS(\gamma)/2 &= \int_{\mathcal{R}} P(S_{k,n,\gamma}(x) \geq 1/2)\left(1 - P(S_{k,n,\gamma}(x) \geq 1/2)\right)dP(x) \\
&= \int_{\mathcal{R}}\left(P(S_{k,n,\gamma}(x) \geq 1/2) - 1_{\{\eta(x)\leq 1/2\}}\right)dP(x) \\
&\quad - \int_{\mathcal{R}}\left(P^2(S_{k,n,\gamma}(x) \geq 1/2) - 1_{\{\eta(x)\leq 1/2\}}\right)dP(x).
\end{aligned}
$$

Based on the definition of $Z_i(x, R_{k+1})$ in (2), the derivation of $\mu_{k,n,\gamma}(x, R_{k+1})$ and $s_{k,n,\gamma}(x, R_{k+1})$, follow the same procedures as in Theorem 5, we obtain

$$
\begin{aligned}
&\int_{\mathcal{R}}\left(P(S_{k,n,\gamma}(x) \geq 1/2) - 1_{\{\eta(x)\leq 1/2\}}\right)dP(x) \\
&= \int_S \int_{-\epsilon_{k,n}}^{\epsilon_{k,n}} f(x_0^t)\left\{\mathbb{E}_{R_{k+1}}P\left(S_{k,n,\gamma}(x_0^t) < 1/2|R_{k+1}\right) - 1_{\{t<0\}}\right\}dtd\text{Vol}^{d-1}(x_0) + o \\
&= \int_S \mathbb{E}_{R_{k+1}}\int_{\mathbb{R}} f(x_0)\left\{\Phi\left(-\frac{t\|\dot\eta(x_0)\|}{\sqrt{s_{k,n,\gamma}^2/k}} - \frac{\mathbb{E}(R_1/R_{k+1})^{-\gamma}a(x_0^t)R_1^2}{\sqrt{s_{k,n,\gamma}^2/k}}\right) - 1_{\{t<0\}}\right\}dtd\text{Vol}^{d-1}(x_0) + o,
\end{aligned}
$$

and similarly,

$$
\begin{aligned}
&\int_{\mathcal{R}}\left(P^2(S_{k,n,\gamma} \geq 1/2|R) - 1_{\{\eta(x)\leq 1/2\}}\right)d\bar{P}(x) \\
&= \int_S \int_{-\epsilon_{k,n}}^{\epsilon_{k,n}} f(x_0^t)\left\{P^2\left(S_{k,n,\gamma}(x_0^t) < 1/2\right) - 1_{\{t<0\}}\right\}dtd\text{Vol}^{d-1}(x_0) + o \\
&= \int_S \mathbb{E}_{R_{k+1}}\int_{\mathbb{R}} f(x_0)\left\{\Phi^2\left(-\frac{t\|\dot\eta(x_0)\|}{\sqrt{s_{k,n,\gamma}^2/k}} - \frac{\mathbb{E}(R_1/R_{k+1})^{-\gamma}a(x_0^t)R_1^2}{\sqrt{s_{k,n,\gamma}^2/k}}\right) - 1_{\{t<0\}}\right\}dtd\text{Vol}^{d-1}(x_0) + o.
\end{aligned}
$$

Adopting Proposition 4 and the fact that $s_{k,n,\gamma}(x, R_{k+1})$ is little changed by $x$ and $R_{k+1}$, treating $\Phi$ and $\Phi^2$ as two distribution functions, we have

$$CIS(\gamma) = \frac{B_1}{\sqrt{\pi}}\frac{1}{\sqrt{k}}\mathbb{E}\sqrt{s_{k,n,\gamma}^2(X)} + o = \frac{B_1}{\sqrt{\pi}}\frac{1}{\sqrt{k}}\sqrt{1 + \frac{\gamma^2}{d(d-2\gamma)}} + o.$$

The optimum $k$ for $k$NN and interpolated-NN satisfies

$$\frac{k(\gamma)}{k(0)} \to \left(\frac{(d+2-\gamma)^2}{d(d-2\gamma)}\frac{d^2}{(d+2)^2}\right)^{\frac{d}{d+4}}.$$

Finally we can obtain the asymptotic ratio of CIS'es.

# F  INTERPOLATION DOES NOT AFFECT THE RATE OF CONVERGENCE

## F.1  MODEL SETUP AND THEOREM

The results in Section 3 are obtained based on smooth $\eta$ and $f$, thus the Taylor expansions can be calculated. In this section, we will provide a weaker result under weaker assumptions as those in Chaudhuri & Dasgupta (2014) and Belkin et al. (2018).

We first setup some assumptions:

A.1'  $X$ is a $d$-dimensional random variable on a compact set and satisfies regularity condition: let $\lambda$ be the Lebesgue measure on $\mathbb{R}^d$, then there exists positive $(c_0, r_0)$ such that for any $x$ in the support $\mathcal{X}$,
$$\lambda(\mathcal{X} \cap B(x, r)) \geq c_0 \lambda(B(x, r)),$$
for any $0 < r \leq r_0$.

A.4'  Smoothness condition: $|\eta(x) - \eta(y)| \leq A\|x - y\|^\alpha$ for some $\alpha > 0$.

A.5'  The density of $X$ is finite, and the model satisfies margin condition: $P(|\eta(X) - 1/2| < t) \leq Bt^\beta$.

The regularity condition in A.1' is used to ensure the convergence of the distance of the $k + 1$th neighbor: if the density is infinite at some points, the neighbors will be the point itself. The smoothness condition A.4' describes how smooth the function $\eta$ is, which affects the minimax performance of $k$NN given a class of $\eta$'s. Assumption A.5' describes how far the samples are away from $1/2$, and affects the minimax rate for binary classification. When $\beta$ is small, there is a cluster of samples around $1/2$, leading to a large prediction error rate.

In interpolated-NN, if the level of over-fitting is not strong, that is, $\gamma$ is within some suitable range, then excess risk of classification also reaches optimal:

**Theorem 6**  *For regression, under A.1', A.3, A.4', A.5',*
$$MSE(\gamma, n) = O(n^{-2\alpha/(2\alpha+d)}).$$
*For classification, under A.1', A.3, A.4', A.5',*
$$Regret(\gamma, n) = O(n^{-\alpha(\beta+1)/(2\alpha+d)}).$$
*In addition, if $\beta \geq 2$, when $d - \kappa(\beta)\gamma > 0$, where $\kappa(\beta)$ denotes the smallest even number that is greater than $\beta + 1$, then we have*
$$Regret(\gamma, n) = O(n^{-\alpha(\beta+1)/(2\alpha+d)}).$$

The main proof of Theorem 6 follows the idea in Chaudhuri & Dasgupta (2014), with Berstein inequality changing to non-uniform Berry Esseen Thoerem. For the regression part, we refer readers to Belkin et al. (2018) and Belkin et al. (2019c).

From Theorem 6, one can see that given a reasonable level of over-fitting, the performance of interpolated-NN reaches the minimax rate for both regression and classification.

When $\beta < 2$ and $\beta \geq 2$, we technically impose different restrictions on $\gamma$. A quick illustration on this is that we adopt non-uniform Berry-Essen Theorem to bound the probability. In non-uniform Berry-Essen Theorem, although the Gaussian probability itself has a exponential tail, the residual part is only $1/(1 + |t|^3)$. As a result, when summing up $t_i^{\beta+1}/(1 + t_i^3)$ for $t_i = 2^i$, $\beta - 2$ should be smaller than 0 in order to bound the summation.

**Remark 3**  *As Chaudhuri & Dasgupta (2014) mentioned, our smoothness condition (A.4') is in fact a sufficient condition. In Theorem 6, what we really need is some constant $\alpha'$ such that*
$$|\eta(x) - \eta(B(x, r))| \leq Lr^{\alpha'}.$$
*The result in Samworth et al. (2012) is a case where $\alpha \neq \alpha'$. We approximate $\mathbb{E}\eta(X)$ as $\eta(x_0) + tr[\ddot{\eta}(x_0)\mathbb{E}(X - x_0)(X - x_0)^\top]$ in Theorem 5, which indicates that we adopt smoothness $\alpha' = 2$, though $\alpha = 1$. In addition, the margin condition parameter $\beta = 1$ under A.1 to A.5 becomes 1. Therefore, the optimum rate of $O(n^{-\alpha'(\beta+1)/(2\alpha'+d)})$ is $O(n^{-4/(4+d)})$. The over-fitting weighting scheme in fact leads to optimal rate for classification.*

## F.2 PROOF OF THEOREM 6

Let $p = 2k/n$. Denote $E = \{\exists R_i > r_p, \ i = 1, \ldots, k\}$, and $\mathbb{E}R_{k,n}(x) - R^*(x)$ as the excess risk. Then define

$$\tilde{\eta}_{k,n,\gamma}(x|R_{k+1}) = \mathbb{E}[(R_1/R_{k+1})^{-\gamma}\eta(X_1)]/\mathbb{E}(R_1/R_{k+1})^{-\gamma},$$

as well as

$$\mathcal{X}^+_{p,\Delta} = \{x \in \mathcal{X} | \eta(x) > \frac{1}{2}, \tilde{\eta}(x) \geq \frac{1}{2} + \Delta, \forall R_{k+1} < r_{2p}(x)\},$$

$$\mathcal{X}^-_{p,\Delta} = \{x \in \mathcal{X} | \eta(x) < \frac{1}{2}, \tilde{\eta}(x) \leq \frac{1}{2} - \Delta, \forall R_{k+1} < r_{2p}(x)\},$$

with the decision boundary area:

$$\partial_{p,\Delta} = \mathcal{X} \setminus (\mathcal{X}^+_{p,\Delta} \cup \mathcal{X}^-_{p,\Delta}).$$

Given $\partial_{p,\Delta}$, $\mathcal{X}^+_{p,\Delta}$, and $\mathcal{X}^-_{p,\Delta}$, similar with Lemma 8 in Chaudhuri & Dasgupta (2014), the event of $g(x) \neq \widehat{g}_{k,n,\gamma}(x)$ can be covered as:

$$
\begin{aligned}
\mathbb{1}_{\{g(x) \neq \widehat{g}_{k,n,\gamma}(x)\}} \quad &\leq \quad \mathbb{1}_{\{x \in \partial_{p,\Delta}\}} \\
&+ \mathbb{1}_{\{\max_{i=1,\ldots,k} R_i \geq r_{2p}\}} \\
&+ \mathbb{1}_{\{|\widehat{\eta}_{k,n,\gamma}(x) - \tilde{\eta}(x|R_{k+1})| \geq \Delta\}}.
\end{aligned}
$$

When $\tilde{\eta}(x|R_{k+1}) > 1/2$ and $x \in \mathcal{X}^+_{p,\Delta}$, assume $\widehat{\eta}_{k,n,\gamma}(x) < 1/2$, then

$$\tilde{\eta}_{k,n,\gamma}(x|R_{k+1}) - \widehat{\eta}_{k,n,\gamma}(x) > \tilde{\eta}_{k,n,\gamma}(x|R_{k+1}) - 1/2 \geq \Delta.$$

The other two events are easy to figure out.

In addition, from the definition of Regret, assume $\eta(x) < 1/2$,

$$
\begin{aligned}
&P(\widehat{g}(x) \neq Y | X = x) - \eta(x) \\
=\quad & \eta(x)P(\widehat{g}(x) = 0 | X = x) + (1 - \eta(x))P(\widehat{g}(x) = 1 | X = x) - \eta(x) \\
=\quad & \eta(x)P(\widehat{g}(x) = g(x) | X = x) + (1 - \eta(x))P(\widehat{g}(x) \neq g(x) | X = x) - \eta(x) \\
=\quad & \eta(x) - \eta(x)P(\widehat{g}(x) \neq g(x) | X = x) + (1 - \eta(x))P(\widehat{g}(x) \neq g(x) | X = x) - \eta(x) \\
=\quad & (1 - 2\eta(x))P(\widehat{g}(x) \neq g(x) | X = x),
\end{aligned}
$$

similarly, when $\eta(x) > 1/2$, we have

$$P(\widehat{g}(x) \neq Y | X = x) - 1 + \eta(x) \quad = \quad (2\eta(x) - 1)P(\widehat{g}(x) \neq g(x) | X = x).$$

As a result, the Regret can be represented as

$$\text{Regret}(k,n,\gamma) \quad = \quad \mathbb{E}\left(|1 - 2\eta(X)|P(g(X) \neq \widehat{g}_{k,n,\gamma}(X))\right).$$

For simplicity, denote $p = k/n$. We then follow the proof of Lemma 20 of Chaudhuri & Dasgupta (2014). Without loss of generality assume $\eta(x) > 1/2$. Define

$$
\begin{aligned}
\Delta_0 \quad &= \quad \sup_x |\tilde{\eta}(x|R_{k+1}) - \eta(x)| = O(r_{2p}^\alpha) = O(k/n)^{\alpha/d}, \\
\Delta(x) \quad &= \quad |\eta(x) - 1/2|,
\end{aligned}
$$

then

$$\tilde{\eta}(x|R_{k+1}) \geq \eta(x) - \Delta_0 = \frac{1}{2} + (\Delta(x) - \Delta_0),$$

hence $x \in \mathcal{X}^+_{p,\Delta(x)-\Delta_0}$.

From the definition of $R_{k,n}$ and $R^*$, when $\Delta(x) > \Delta_0$, we have

$$\mathbb{E}R_{k,n}(x) - R^*(x)$$

$$\leq \quad 2\Delta(x)\left[P(r_{(k+1)} > v_{2p}) + P\left(\sum_{i=1}^{k} W_i Y(X_i) - \tilde{\eta}(x|R_{k+1}) > \Delta(x) - \Delta_0\right)\right]$$

$$\leq \quad \exp(-k/8) + 2\Delta(x)P\left(\sum_{i=1}^{k}(R_i/R_{k+1})^{-\gamma}Y(X_i) > (\tilde{\eta}(x|R_{k+1}) + \Delta(x) - \Delta_0)\sum_{i=1}^{k}(R_i/R_{k+1})^{-\gamma}\right)$$

$$= \quad \exp(-k/8) + 2\Delta(x)P\left(\sum_{i=1}^{k}(R_i/R_{k+1})^{-\gamma}Y(X_i) - k\mathbb{E}(R_1/R_{k+1})^{-\gamma}\eta(X_1)\right.$$

$$-(\tilde{\eta}(x|R_{k+1}) + \Delta(x) - \Delta_0)\sum_{i=1}^{k}\left[(R_i/R_{k+1})^{-\gamma} - \mathbb{E}(R_1/R_{k+1})^{-\gamma}\right]$$

$$\left. > k(\tilde{\eta}(x|R_{k+1}) + \Delta(x) - \Delta_0)\mathbb{E}(R_1/R_{k+1})^{-\gamma} - k\mathbb{E}(R_1/R_{k+1})^{-\gamma}\eta(X_1)\right)$$

$$= \quad \exp(-k/8) + 2\Delta(x)P\left\{\sum_{i=1}^{k}(R_i/R_{k+1})^{-\gamma}(Y(X_i) - \tilde{\eta}(x|R_{k+1}))\right.$$

$$-(\Delta(x) - \Delta_0)\left(\sum_{i=1}^{k}(R_i/R_{k+1})^{-\gamma} - k\mathbb{E}(R_1/R_{k+1})^{-\gamma}\right)$$

$$\left. > k(\Delta(x) - \Delta_0)\mathbb{E}(R_1/R_{k+1})^{-\gamma}\right\}.$$

Since

$$\mathbb{E}\sum_{i=1}^{k}(R_i/R_{k+1})^{-\gamma}(Y(X_i) - \tilde{\eta}(x|R_{k+1})) = 0,$$

$$\mathbb{E}(\Delta(x) - \Delta_0)\left(\sum_{i=1}^{k}(R_i/R_{k+1})^{-\gamma} - k\mathbb{E}(R_1/R_{k+1})^{-\gamma}\right) = 0,$$

we can use Markov inequality to the power of $\kappa(\beta)$ to bound the probability. Denote

$$Z_i(x) = (R_i/R_{k+1})^{-\gamma}(Y(X_i) - \tilde{\eta}(x|R_{k+1})) - (\Delta(x) - \Delta_0)(R_i/R_{k+1})^{-\gamma} + (\Delta(x) - \Delta_0)\mathbb{E}(R_1/R_{k+1})^{-\gamma}$$

for simplicity. Note that

$$Var(Z_1(x)) \quad = \quad O(\Delta(x) - \Delta_0).$$

For different settings of $\beta$ and $\gamma$, the following steps have the same logic but different details:

**Case 1: $\beta \leq 1$ and $\gamma < d/3$:** Considering the problem that the upper bound can be much greater than 1 when $\Delta(x)$ is small, we define $\Delta_i = 2^i \Delta_0$, taking $i_0 = \min\{i \geq 1 | (\Delta_i - \Delta_0)^2 > 1/k\}$. In this situation, since $\mathbb{E}Z_1^3(x) < \infty$, we can adopt non-uniform Berry-Essen Theorem for the proof:

$$\mathbb{E}R_{k,n}(X) - R^*(X) \quad = \quad \mathbb{E}(R_{k,n}(X) - R^*(X))1_{\{\Delta(X) \leq \Delta_{i_0}\}}$$

$$+\mathbb{E}(R_{k,n}(X) - R^*(X))1_{\{\Delta(X) > \Delta_{i_0}\}}$$

$$\leq \quad 2\Delta_{i_0}P(\Delta(X) \leq \Delta_{i_0}) + \exp(-k/8)$$

$$+4\mathbb{E}\left[\Delta(X)1_{\{\Delta_{i_0} < \Delta(X)\}}\bar{\Phi}\left(\frac{\sqrt{k}(\Delta(x) - \Delta_0)}{\sqrt{Var(Z_1(X)|X)}}\right)\right]$$

$$+4\mathbb{E}\left[\Delta(X)1_{\{\Delta_{i_0} < \Delta(X)\}}\frac{c_1}{\sqrt{k}}\frac{1}{1 + k^{3/2}(\Delta(x) - \Delta_0)^3}\right].$$

The two terms from Berry-Essen Theorem becomes

$$\mathbb{E}\left[\Delta(X)1_{\{\Delta_i<\Delta(X)<\Delta_{i+1}\}}\frac{c_1}{\sqrt{k}}\frac{1}{1+k^{3/2}(\Delta(x)-\Delta_0)^3}\right]$$

$$\leq \quad \Delta_{i+1}^{\beta+1}\frac{c_1}{\sqrt{k}}\frac{1}{1+k^{3/2}(\Delta_i-\Delta_0)^3}$$

$$= \quad \frac{1}{k^{(\beta+1)/2}}(\sqrt{k}\Delta_{i+1})^{\beta+1}\frac{c_1}{\sqrt{k}}\frac{1}{1+k^{3/2}(\Delta_i-\Delta_0)^3},$$

together with

$$\mathbb{E}\left[\Delta(X)1_{\{\Delta_{i_0}<\Delta(X)\}}\bar{\Phi}\left(\frac{\sqrt{k}(\Delta(x)-\Delta_0)}{\sqrt{Var(Z_1(X)|X)}}\right)\right]$$

$$\leq \quad \mathbb{E}\left[\Delta(X)1_{\{\Delta_{i_0}<\Delta(X)\}}\bar{\Phi}\left(c_3\sqrt{k}(\Delta(x)-\Delta_0)\right)\right]$$

$$\leq \quad c_4\frac{\Delta_{i+1}^{\beta+1}}{\sqrt{k}(\Delta_i-\Delta_0)}\exp\left(-c_3^2k(\Delta_i-\Delta_0)^2\right).$$

The upper bound is larger than 1 if $\Delta_i \leq c_5/\sqrt{k}$. When $\Delta_i > c_5/\sqrt{k}$,

$$\frac{\frac{\Delta_{i+1}^{\beta+1}}{(\Delta_i-\Delta_0)}\exp\left(-c_3^2k(\Delta_i-\Delta_0)^2\right)}{\frac{\Delta_i^{\beta+1}}{(\Delta_{i-1}-\Delta_0)}\exp\left(-c_3^2k(\Delta_{i-1}-\Delta_0)^2\right)} \quad = \quad 2^{\beta+1}\frac{2^{i-1}-1}{2^i-1}\frac{\exp\left(-c_3^2k(\Delta_i-\Delta_0)^2\right)}{\exp\left(-c_3^2k(\Delta_{i-1}-\Delta_0)^2\right)}$$

$$\leq \quad 2^\beta\frac{\exp\left(-c_3^2k(\Delta_i-\Delta_0)^2\right)}{\exp\left(-c_3^2k(\Delta_{i-1}-\Delta_0)^2\right)} < 1/2.$$

Therefore the sum of the excess risk can be bounded. When $\beta < 2$, $\beta - 2 < 0$, hence

$$\mathbb{E}\left[\Delta(X)1_{\{\Delta_i<\Delta(X)<\Delta_{i+1}\}}\frac{c_1}{\sqrt{k}}\frac{1}{1+k^{3/2}(\Delta(x)-\Delta_0)^3}\right]$$

$$\leq \quad O\left(\frac{1}{k^{(\beta+2)/2}}\right)\sum_{i\geq i_0}[\sqrt{k}(\Delta_{i+1}-\Delta_i)]^{\beta-2}$$

$$\leq \quad O\left(\frac{1}{k^{(\beta+2)/2}}\right)\sum_{i\geq i_0}(\sqrt{k}\Delta_{i_0})^{\beta-2}2^{i(\beta-2)}$$

$$= \quad O\left(\frac{1}{k^{(\beta+2)/2}}k^{(\beta-2)/2}\Delta_{i_0}^{\beta-2}\right) = O\left(\frac{\Delta_{i_0}^{\beta-2}}{k^2}\right),$$

and

$$\mathbb{E}\left[\Delta(X)1_{\{\Delta_{i_0}<\Delta(X)\}}\bar{\Phi}\left(\frac{\sqrt{k}(\Delta(x)-\Delta_0)}{\sqrt{Var(Z_1(X)|X)}}\right)\right]$$

$$\leq \quad O\left(\frac{1}{\sqrt{k}}\right)\sum_{i\geq i_0}\frac{\Delta_{i+1}^{\beta+1}}{(\Delta_i-\Delta_0)}\exp\left(-c_3^2k(\Delta_i-\Delta_0)^2\right)$$

$$= \quad O\left(\frac{1}{\sqrt{k}}\right)\Delta_{i_0+1}^\beta\exp\left(-c_3^2k(\Delta_{i_0}-\Delta_0)^2\right).$$

Recall that $\Delta_{i_0} > \Delta_0$ and $\Delta_{i_0}^2 > 1/k$, hence when $\Delta_{i_0}^2 = O(1/k)$, we can obtain the minimum upper bound

$$\mathbb{E}R_{k,n}(X) - R^*(X) \leq O(\Delta_0^{\beta+1}) + O\left(\left(\frac{1}{k}\right)^{(\beta+1)/2}\right).$$

Taking $k \asymp (n^{2\alpha/(2\alpha+d)})$, the upper bound becomes $O(n^{-\alpha(\beta+1)/(2\alpha+d)})$.

**Case 2:** $\gamma < d/\kappa(\beta)$:

$$
\begin{aligned}
\mathbb{E}R_{k,n}(X) - R^*(X) &= \mathbb{E}(R_{k,n}(X) - R^*(X))1_{\{\Delta(X)\le\Delta_{i_0}\}} \\
&\quad + \mathbb{E}(R_{k,n}(X) - R^*(X))1_{\{\Delta(X)>\Delta_{i_0}\}} \\
&\le 2\Delta_{i_0}P(\Delta(X) \le \Delta_{i_0}) + \exp(-k/8) \\
&\quad + 4\mathbb{E}\left[\Delta(X)1_{\{\Delta_{i_0}<\Delta(X)\}}\frac{\mathbb{E}\left(\sum_{i=1}^k Z_i(X)\right)^{\kappa(\beta)}}{(\Delta(X)-\Delta_0)^{\kappa(\beta)}k^{\kappa(\beta)}\mathbb{E}^{\kappa(\beta)}(R_1/R_{k+1})^{-\gamma}}\right],
\end{aligned}
$$

while for some constant $c_1 > 0$,

$$
\mathbb{E}\left[\Delta(X)1_{\{\Delta_i<\Delta(X)\le\Delta_{i+1}\}}\frac{\mathbb{E}\left(\sum_{i=1}^k Z_i(X)\right)^{\kappa(\beta)}}{(\Delta(X)-\Delta_0)^{\kappa(\beta)}k^{\kappa(\beta)}\mathbb{E}^{\kappa(\beta)}(R_1/R_{k+1})^{-\gamma}}\right]
$$

$$
\begin{aligned}
&\le \mathbb{E}\left[\Delta(X)1_{\{\Delta_i<\Delta(X)\le\Delta_{i+1}\}}\right]\frac{\mathbb{E}\left(\sum_{i=1}^k Z_i(X)\right)^{\kappa(\beta)}}{(\Delta_i-\Delta_0)^{\kappa(\beta)}k^{\kappa(\beta)}\mathbb{E}^{\kappa(\beta)}(R_1/R_{k+1})^{-\gamma}} \\
&\le \Delta_{i+1}P(\Delta(X)\le\Delta_{i+1})\frac{\mathbb{E}\left(\sum_{i=1}^k Z_i(X)\right)^{\kappa(\beta)}}{(\Delta_i-\Delta_0)^{\kappa(\beta)}k^{\kappa(\beta)}\mathbb{E}^{\kappa(\beta)}(R_1/R_{k+1})^{-\gamma}} \\
&\le c_1\left(\frac{1}{k}\right)^{\kappa(\beta)/2}\Delta_{i+1}^{\beta+1}/(\Delta_i-\Delta_0)^{\kappa(\beta)},
\end{aligned}
$$

where the last inequality is obtained since $d - \kappa(\beta)\gamma > 0$. Note that $\kappa(\beta) > \beta + 1$, thus for some $0 < c < 1$,

$$
\frac{\Delta_{i+1}^{\beta+1}/(\Delta_i-\Delta_0)^{\kappa(\beta)}}{\Delta_i^{\beta+1}/(\Delta_{i-1}-\Delta_0)^{\kappa(\beta)}} < c. \tag{5}
$$

Therefore the sum of the excess risk for can be bounded, where

$$
\mathbb{E}\left[\Delta(X)1_{\{\Delta_{i_0}<\Delta(X)\}}\frac{\mathbb{E}\left(\sum_{i=1}^k Z_i(X)\right)^{\kappa(\beta)}}{(\Delta(X)-\Delta_0)^{\kappa(\beta)}k^{\kappa(\beta)}\mathbb{E}^{\kappa(\beta)}(R_1/R_{k+1})^{-\gamma}}\right]
$$

$$
\begin{aligned}
&\le O\left(\frac{1}{k}\right)^{\kappa(\beta)/2}\sum_{i\ge i_0}\Delta_{i+1}^{\beta+1}/(\Delta_i-\Delta_0)^{\kappa(\beta)} \tag{6} \\
&\le O\left(\Delta_{i_0}^{\beta+1}\right).
\end{aligned}
$$

Recall that $\Delta_{i_0} > \Delta_0$ and $\Delta_{i_0}^2 > 1/k$, hence when $\Delta_{i_0}^2 = O(1/k)$, we can obtain the minimum upper bound

$$
\mathbb{E}R_{k,n}(X) - R^*(X) \le O(\Delta_0^{\beta+1}) + O\left(\left(\frac{1}{k}\right)^{(\beta+1)/2}\right).
$$

Taking $k \asymp (n^{2\alpha/(2\alpha+d)})$, the upper bound becomes $O(n^{-\alpha(\beta+1)/(2\alpha+d)})$.

