# OpenReview forum: "Benefit of Interpolation in Nearest Neighbor Algorithms"
_ICLR.cc/2020/Conference — Reject_

### Official Review · AnonReviewer1 · 2019-10-24
**Official Blind Review #1**

**Rating:** 1

**Review:**

This paper is about interpolation schemes in the particular case of the
Nearest Neighbor algorithm. The authors investigate the benet, mainly
theoretical, of the proposed interpolation. They study minimax rates of
the proposed interpolated-NN for both classication and regression. The
statistical stability of the Interpolated-NN is adressed.
The paper is easy to understand and correctly written. Nevertheless,
It is a particular case of ( \Overtting or perfect tting? Risk bounds for
classication and regression rules that interpolate", Belkin, M., Hsu, D.,
and Mitra, P. (2018a)) with an explicit interpolation schemes given by the
euclidien distance power gamma. It appears as an application of the above
paper which brings only few theoretical advantages and not enough to justify,
although intuitive, the choice of these weights. Only few discussions and
no comparison to others bounds (as those in the above paper) of the main
theorem are given. The paper is too much incremental from the papers of
Belkin et al. (2018) and Xing et al. (2018) and the benets of the proposed
interpolation are limited. Furthermore, the empirical performance of the
interpolate-NN is clearly not convincing and show no signicant practical
advantages of the proposed method. As the goal of the paper is clearly
theoretical, the 'real data analysis' part is not necessary in my opinion.

**Experience Assessment:**

I have published in this field for several years.

**Review Assessment: Checking Correctness Of Derivations And Theory:**

I carefully checked the derivations and theory.

**Review Assessment: Checking Correctness Of Experiments:**

I carefully checked the experiments.

**Review Assessment: Thoroughness In Paper Reading:**

I read the paper thoroughly.

---

> ### Author Response · Authors · 2019-11-15
> **To Reviewer 1**
>
> In Belkin (2018), they technically only obtains a suboptimal bound for classification. And from their theorem, even if it is an optimal rate, it is only sufficient to state that "interpolated-NN is not hurt by interpolation". Our work, by proving that interpolated NN yields a smaller multiplicative constant, asserts that "interpolation helps improve the performance".

---

> ### Author Response · Authors · 2019-11-15
> **To Reviewer 1&3**
>
> The goal of this work is not to pursue the rate of convergence of interpolated-NN (which has been done in Belkin (2018) and Xing (2019). ) Our main theorem provides the EXACT MSE and Regret, rather than rate result up to unknown multiplicative constants. Therefore, it is not a comparable result to Belkin (2018) and Xing (2019), but a sharper improvement. Traditional kNN, interpolated-NN and the OWNN (Samworth, 2012) are all rate optimal, but our result enable us to rigorously compare the three on the multiplicative constant level (as described in Section 3.4).
>
> The study of multiplicative constant, beyond the rate of convergence, provides more subtle insights. For example, Samworth (2012) claimed that the OWNN is the optimal weight choice by proving its multiplicative constant is the smallest; Locally-weighted NN (Cannings, et al, 2019) calculated the exact Regret to prove that pointwise local weighting scheme is better than a uniform choice of k.  More reference of studies on multiplicative constant is provided in the reference list.
>
> Our work, beyond the results of Belkin (2018) and Xing (2019), characterizes how the performance of interpolated-NN changes with respect to the interpolated level (i.e., gamma) and reveals a “double descent” phenomenon in NN algorithm which echoes many recent studies for over-parametrized models. This new insight is the most important message we would like to deliver to the readers, rather than the convergence rate results. Another insight is that a proper interpolation may be viewed as a form of regularization (in terms of slightly reducing bias) that improves the predictive power of optimal kNN.
>
>  Reference:
> [1] Cannings, T. I., Berrett, T. B. and Samworth, R. J. (2019) Local nearest neighbour classification with applications to semi-supervised learning. Ann. Statist., to appear.
> [2] Cannings, T. I., Fan, Y. and Samworth, R. J. (2019) Classification with imperfect training labels. Biometrika, to appear.
> [3] Samworth, R. J. (2012) Optimal weighted nearest neighbour classifiers. Ann. Statist., 40, 2733-2763. DOI: 10.1214/12-AOS1049.
> [4] Sun, Will Wei, Xingye Qiao, and Guang Cheng. "Stabilized nearest neighbor classifier and its statistical properties." Journal of the American Statistical Association 111.515 (2016): 1254-1265.
> [5] Duan, Jiexin, Xingye Qiao, and Guang Cheng. "Distributed Nearest Neighbor Classification." arXiv preprint arXiv:1812.05005 (2018).

---

### Official Review · AnonReviewer3 · 2019-10-24
**Official Blind Review #3**

**Rating:** 3

**Review:**

This paper studies the interpolated k-nearest neighbors algorithm from a theoretical perspective. Specifically, it studies how the performance of the algorithm is affected by reweighting the k nearest neighbors according to their relative distance. This regime has been considered in prior work, particularly Belkin et al. (2018).

Under various niceness conditions, the paper proves error bounds for interpolated k-nearest neighbors for both regression (i.e. squared loss) and classification (i.e. 0-1 loss after thresholding).

Overall, I have the impression that this paper contains interesting ideas, but the presentation is very poor. It should be revised and resubmitted before it can be accepted.

In particular, the paper does not make its contribution clear. The main theorem only appears on page 4 and the reader must consult the appendix to see the definition of all the terms that appear in the theorem. I have no idea how to interpret the (complicated) expression in the theorem. The theorem needs to be explained in intuitive terms. More context needs to be given by comparing the main theorem to prior works (which I am not familiar with).




**Experience Assessment:**

I have read many papers in this area.

**Review Assessment: Checking Correctness Of Derivations And Theory:**

I assessed the sensibility of the derivations and theory.

**Review Assessment: Checking Correctness Of Experiments:**

I did not assess the experiments.

**Review Assessment: Thoroughness In Paper Reading:**

I read the paper at least twice and used my best judgement in assessing the paper.

---

> ### Author Response · Authors · 2019-11-15
> **To Reviewer 2&3**
>
> Thanks for pointing out our writing problem.
>
> The theorem 1 in the first submission shall serve as an important intermediate (i.e lemma) result. The corollaries derived from theorem 1 is actually our main point. We adjusted the displays of main theorems in a revision submission to enhance the readability.

---

> ### Author Response · Authors · 2019-11-15
> **To Reviewer 1&3**
>
> The goal of this work is not to pursue the rate of convergence of interpolated-NN (which has been done in Belkin (2018) and Xing (2019). ) Our main theorem provides the EXACT MSE and Regret, rather than rate result up to unknown multiplicative constants. Therefore, it is not a comparable result to Belkin (2018) and Xing (2019), but a sharper improvement. Traditional kNN, interpolated-NN and the OWNN (Samworth, 2012) are all rate optimal, but our result enable us to rigorously compare the three on the multiplicative constant level (as described in Section 3.4).
>
> The study of multiplicative constant, beyond the rate of convergence, provides more subtle insights. For example, Samworth (2012) claimed that the OWNN is the optimal weight choice by proving its multiplicative constant is the smallest; Locally-weighted NN (Cannings, et al, 2019) calculated the exact Regret to prove that pointwise local weighting scheme is better than a uniform choice of k.  More reference of studies on multiplicative constant is provided in the reference list.
>
> Our work, beyond the results of Belkin (2018) and Xing (2019), characterizes how the performance of interpolated-NN changes with respect to the interpolated level (i.e., gamma) and reveals a “double descent” phenomenon in NN algorithm which echoes many recent studies for over-parametrized models. This new insight is the most important message we would like to deliver to the readers, rather than the convergence rate results. Another insight is that a proper interpolation may be viewed as a form of regularization (in terms of slightly reducing bias) that improves the predictive power of optimal kNN.
>
>  Reference:
> [1] Cannings, T. I., Berrett, T. B. and Samworth, R. J. (2019) Local nearest neighbour classification with applications to semi-supervised learning. Ann. Statist., to appear.
> [2] Cannings, T. I., Fan, Y. and Samworth, R. J. (2019) Classification with imperfect training labels. Biometrika, to appear.
> [3] Samworth, R. J. (2012) Optimal weighted nearest neighbour classifiers. Ann. Statist., 40, 2733-2763. DOI: 10.1214/12-AOS1049.
> [4] Sun, Will Wei, Xingye Qiao, and Guang Cheng. "Stabilized nearest neighbor classifier and its statistical properties." Journal of the American Statistical Association 111.515 (2016): 1254-1265.
> [5] Duan, Jiexin, Xingye Qiao, and Guang Cheng. "Distributed Nearest Neighbor Classification." arXiv preprint arXiv:1812.05005 (2018)

---

### Official Review · AnonReviewer2 · 2019-11-08
**Official Blind Review #2**

**Rating:** 6

**Review:**

The paper studies theoretical perspective of double descent phenomenon for the interpolated K-NN classifier.

The paper is works in several interesting directions and gives theoretical reasoning to how interpolated K-NN could exhibit the double descent phenomenon. They give theoretical justifications albeit with strong assumptions.

I think the paper is a good paper. However, I have concerns with the presentation quality of the paper. It is very tough to get through the paper till the end.

In my view, it would have been an Accept if the paper was well written.

**Experience Assessment:**

I have read many papers in this area.

**Review Assessment: Checking Correctness Of Derivations And Theory:**

I assessed the sensibility of the derivations and theory.

**Review Assessment: Checking Correctness Of Experiments:**

I assessed the sensibility of the experiments.

**Review Assessment: Thoroughness In Paper Reading:**

I read the paper at least twice and used my best judgement in assessing the paper.

---

> ### Author Response · Authors · 2019-11-15
> **To Reviewer 2&3**
>
> Thanks for pointing out our writing problem.
>
> The theorem 1 in the first submission shall serve as an important intermediate (i.e lemma) result. The corollaries derived from theorem 1 is actually our main point. We adjusted the displays of main theorems in a revision submission to enhance the readability.

---

### Decision · Program_Chairs · 2019-12-19

**Decision:**

Reject

**Comment:**

The authors show that data interpolation in the context of nearest neighbor algorithms, can sometime strictly improve performance. The paper is poorly written for an ICLR audience and the added value compared to extensive prior work in the area is not clearly demonstrated.